# BAYES RISK CTC: CONTROLLABLE CTC ALIGNMENT IN SEQUENCE-TO-SEQUENCE TASKS

**Jinchuan Tian, Jianwei Yu**[*]**, Chao Weng, Dong Yu**
Tencent AI LAB
{tyriontian, tomasyu, cweng, dyu}@tencent.com

**Brian Yan & Shinji Watanabe**[*]
Language Technologies Institute, Carnegie Mellon University, Pittsburgh, PA 15213, USA
{byan, swatanab}@andrew.cmu.edu

## ABSTRACT

Sequence-to-Sequence (seq2seq) tasks transcribe the input sequence to a target sequence. The Connectionist Temporal Classification (CTC) criterion is widely used in multiple seq2seq tasks. Besides predicting the target sequence, a side product of CTC is to predict the alignment, which is the most probable input-long sequence that specifies a hard aligning relationship between the input and target units. As there are multiple potential aligning sequences (called *paths*) that are equally considered in CTC formulation, the choice of which path will be most probable and become the predicted alignment is always uncertain. In addition, it is usually observed that the alignment predicted by vanilla CTC will drift compared with its reference and rarely provides practical functionalities. Thus, the motivation of this work is to make the CTC alignment prediction controllable and thus equip CTC with extra functionalities. The Bayes risk CTC (BRCTC) criterion is then proposed in this work, in which a customizable Bayes risk function is adopted to enforce the desired characteristics of the predicted alignment. With the risk function, the BRCTC is a general framework to adopt some customizable preference over the paths in order to concentrate the posterior into a particular subset of the paths. In applications, we explore one particular preference which yields models with the down-sampling ability and reduced inference costs. By using BRCTC with another preference for early emissions, we obtain an improved performance-latency trade-off for online models. Experimentally, the proposed BRCTC, along with a trimming approach, enables us to reduce the inference cost of offline models by up to 47% without performance degradation; BRCTC also cuts down the overall latency of online systems to an unseen level[1].

## 1 INTRODUCTION

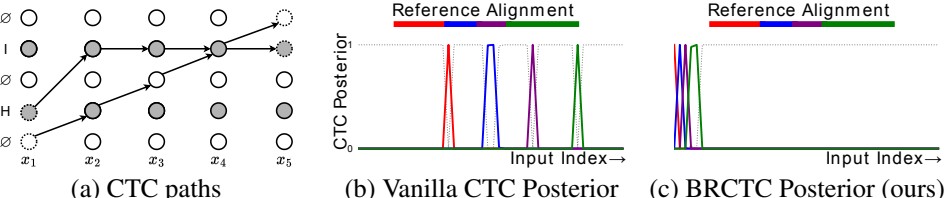

(a) CTC paths     (b) Vanilla CTC Posterior     (c) BRCTC Posterior (ours)

Figure 1: (a) An intuitive explanation of CTC paths. $\varnothing$ is the blank symbol. Each path suggests a hard alignment between the input and target. (b) Posterior of an offline vanilla CTC ASR system. Different colors mean different units. The predicted alignment drifts away from its reference[2] but the predicted non-blank token sequence is correct. (c) Posterior of a BRCTC ASR system that adopts the method in section 3.3. All non-blank spikes are squeezed to the earlier time stamps.

Sequence-to-Sequence (seq2seq) tasks have attracted broad interest and achieved great progress in multiple applications in the past few decades. Connectionist Temporal Classification (CTC) (Graves et al., 2006) is a fundamental criterion for seq2seq tasks. The CTC criterion was initially proposed

---

[1]Code release: https://github.com/espnet/espnet. * means corresponding authors.
[2]Reference alignment is obtained by a deep neural network-hidden Markov model (DNN-HMM) system.

for automatic speech recognition (ASR) but its usage has been extended to many other tasks like machine translation (MT) (Qian et al., 2021; Gu & Kong, 2020; Huang et al., 2022), speech translation (ST) (Yan et al., 2022; Chuang et al., 2021; Liu et al., 2020), sign language translation (Wang et al., 2018; Guo et al., 2019; Camgoz et al., 2020), optical character recognition (OCR) (Graves & Schmidhuber, 2008), lip reading (Assael et al., 2017), hand gesture detection (Molchanov et al., 2016) and even robot control (Shiarlis et al., 2018). Research on CTC is of wide interest, as many advanced systems for seq2seq tasks are based on CTC (Yao et al., 2021), its extensions (Graves, 2012; Sak et al., 2017; Higuchi et al., 2020; Qian et al., 2021) and its hybrid with attention-based architectures (Watanabe et al., 2017; Yan et al., 2022).

In CTC, each input unit is explicitly aligned to either a target unit or a blank symbol. During training, all of these potential aligning sequences (called *paths*) are enumerated and their posteriors are summed and maximized, which is equivalent to maximizing the posterior of the target sequence. Fig.1.a gives an explanation of the paths in CTC. Besides predicting the target sequence, another functionality of CTC is to predict the input-target *alignment*. Unlike the attention-based methods (Chan et al., 2016; Vaswani et al., 2017) that softly predict the aligning relationship by attention weights, CTC predicts a hard alignment. Usually, there is a path whose posterior is dominantly larger than the others (Zeyer et al., 2021), so this dominant path is considered the predicted hard alignment between the input and the target sequences. In CTC implementation, unit-level classification over all possible target units is conducted for each input unit to obtain the posterior of each path. Fig.1.b demonstrates the dominant posterior of the predicted alignment by plotting the unit-level posteriors.

Since predicting any path will yield the correct target sequence, the vanilla CTC is designed to treat all paths equally. However, this equality for paths will result in uncertainty about which path will be selected as the predicted alignment. Also, both our experiments (see Appendix K) and literature (Sak et al., 2015) show that there is a disagreement between the predicted alignment and its reference (see Fig.1.b), which limits its usage in real applications. Thus, the motivation of this work is to control the CTC alignment prediction, making it certain and functional. Specifically, instead of pursuing the accuracy of alignment prediction, e.g., for CTC segmentation (Kürzinger et al., 2020), this work intentionally selects the path with customizable characteristics as the predicted alignment.

This paper proposes a novel Bayes risk CTC (BRCTC) criterion to make CTC alignment prediction controllable. To express our preference for the paths with the desired characteristics, a Bayes risk function is adopted to weigh all paths during training. To be more detailed, the forward-backward algorithm of the original CTC is revised into a divide-and-conquer manner: the paths are firstly divided into several exclusive groups according to a customizable property, and the path groups with more preferred property will receive larger risk values during training. Same as the vanilla CTC, BRCTC can preserve the models' transcription ability, as it considers the posterior of all paths during training. However, the alignment prediction from BRCTC will additionally obtain the desired characteristics due to the adoption of the risk function. Note the designs of how the paths are grouped and what risk value is assigned to each path group are customizable, so the exact functionalities can be tailor-made according to specific applications, such as offline and online scenarios.

In applications, the BRCTC provides novel solutions to two key problems of seq2seq tasks. For offline systems, BRCTC can help to down-sample the intermediate hidden representations so that the mismatch between the input and target lengths is alleviated and the inference cost is significantly reduced. For online systems like streaming ASR, BRCTC provides a better trade-off between transcription quality and latency. Besides ASR, the proposed BRCTC criterion can also be generalized to other seq2seq tasks like MT and ST. Experimentally, BRCTC can cooperate with a trimming approach to achieve up to 47% inference cost reduction for offline systems without degradation in transcription performance; it can also build online systems with extremely low overall latency that can hardly be achieved by vanilla CTC.

Our main contributions are listed as follows: (1) Bayes risk CTC (BRCTC), an extension of CTC, is proposed as a customizable approach to achieve controllable CTC alignment prediction. To the best of our knowledge, this is among the first works which achieve alignment control for CTC-based models without external information. (2) With various intentional designs of the risk functions, BRCTC can significantly reduce the inference cost (by up to 47% relative) and overall latency (to 302ms, up to 30% relative) for offline and online models respectively. (3) Strong experimental evidence is provided in this work to show that high-quality CTC target predictions can be obtained from CTC / BRCTC posteriors which do not necessarily encode accurate alignment prediction.

## 2 REVIEW ON CONNECTIONIST TEMPORAL CLASSIFICATION (CTC)

Seq2seq tasks are to transcribe the input sequence $\mathbf{x} = [\mathbf{x}_1, ..., \mathbf{x}_T]$ to the target sequence $\mathbf{l} = [l_1, ..., l_U]$, where any $\mathbf{x}_t$ is a vector (e.g., features or token embeddings) while any token $l_u$ belongs to a known vocabulary $\mathcal{L}$. $T$ and $U$ are the lengths of input and target respectively. Unless other specified, our discussion is temporarily restricted to ASR. Generalizing the concept of CTC to MT and ST tasks needs more discussion, which is presented in section 5.

### 2.1 TRAINING PROCESS AND ALIGNMENT PREDICTION

CTC (Graves et al., 2006) is a widely used criterion in seq2seq tasks. Following the Bayesian decision theory, CTC tries to maximize the posterior $P(\mathbf{l}|\mathbf{x})$ during training. Instead of maximizing it directly, CTC maximizes the summed probability of all *paths* (see Fig.1.a). Note $\varnothing$ as the blank symbol and extend the vocabulary $\mathcal{L}' = \mathcal{L} \cup \{\varnothing\}$. Any symbol sequence $\pi = [\pi_1, ..., \pi_T] \in \mathcal{L}'^T$ is a path if $\mathcal{B}(\pi) = \mathbf{l}$, where $\mathcal{B}$ is a deterministic mapping that removes all $\varnothing$ and repetitive tokens but preserves the repetitive tokens separated by $\varnothing$ (e.g., $\mathcal{B}(\varnothing aa\varnothing abb) = aab$). Thus, to maximize the posterior $P(\mathbf{l}|\mathbf{x})$ is equivalent to maximizing the summed posterior of all paths:

$$P(\mathbf{l}|\mathbf{x}) = \sum_{\pi \in \mathcal{B}^{-1}(\mathbf{l})} p(\pi|\mathbf{x}) \tag{1}$$

where $\mathcal{B}^{-1}(\mathbf{l})$ is the set of all paths. Next, to compute the posterior $p(\pi|\mathbf{x})$ of each path $\pi$, unit-level posterior over $\mathcal{L}'$ is computed for each input unit, which yields the CTC posterior $\mathbf{y} = [\mathbf{y}^1, ..., \mathbf{y}^T]$. Here each element $\mathbf{y}^t$ is a distribution over $\mathcal{L}'$ at step $t$ and $y_{\pi_t}^t$ is the posterior for element $\pi_t$. So the path posterior is formulated as:

$$p(\pi|\mathbf{x}) = \prod_{t=1}^{T} p(\pi_t|\pi_{1:t-1}, \mathbf{x}) \approx \prod_{t=1}^{T} p(\pi_t|\mathbf{x}) = \prod_{t=1}^{T} y_{\pi_t}^t \tag{2}$$

where the context $\pi_{1:t-1}$ is discarded in the approximation concerning the *conditional independence assumption* of CTC.

As demonstrated in Fig.1.a, any path represents an aligning relationship between the input sequence $\mathbf{l}$ and the target sequence $\mathbf{x}$. Commonly, the CTC posterior is peaky (Zeyer et al., 2021) and the posterior of one certain path is dominantly larger than all others. To this end, the path with the highest posterior is usually considered the predicted alignment during training: $\text{ali}(\mathbf{l}, \mathbf{x}) = \arg\max_{\pi \in \mathcal{B}^{-1}(\mathbf{l})} p(\pi|\mathbf{x})$.

### 2.2 FORWARD-BACKWARD ALGORITHM

Since the number of possible paths in $\mathcal{B}^{-1}(\mathbf{l})$ will grow exponentially with $T$ and $U$ increasing, directly enumerating all paths and their corresponding posteriors is impractical. As an alternative, the *forward-backward algorithm* enables the training objective of CTC to be computed efficiently.

The first step of the forward-backward algorithm is to extend the label sequence $\mathbf{l} = [l_1, ..., l_U]$ into an extended sequence $\mathbf{l}' = [\varnothing, l_1, ..., \varnothing, l_U, \varnothing]$ by inserting a $\varnothing$ between every two non-blank tokens as well as the start and the end of the sequence so that $|\mathbf{l}'| = 2U + 1$[3]. Then, define the forward variable $\alpha(t, v), (1 \leq t \leq T, 1 \leq v \leq 2U + 1)$ as the summed posterior of all path prefix $\pi_{1:t}$ that are aligned with the prefix of the expanded sequence $\mathbf{l}'_{1:v}$; symmetrically, define the backward variable $\beta(t, v), (1 \leq t \leq T, 1 \leq v \leq 2U + 1)$ as the summed posterior of all path suffix $\pi_{t:T}$ that are aligned with the suffix of the expanded sequence $\mathbf{l}'_{v:2U+1}$:

$$\alpha(t, v) = \sum_{\substack{\pi:\mathcal{B}(\pi_{1:t})=\mathcal{B}(\mathbf{l}'_{1:v}) \\ \pi_t = l'_v}} \prod_{t'=1}^{t} y_{\pi_{t'}}^{t'}, \qquad \beta(t, v) = \sum_{\substack{\pi:\mathcal{B}(\pi_{t:T})=\mathcal{B}(\mathbf{l}'_{v:2U+1}) \\ \pi_t = l'_v}} \prod_{t'=t}^{T} y_{\pi_{t'}}^{t'} \tag{3}$$

With any fixed $t$ and $v$, the summed probability of all paths whose $t$-th elements $\pi_t$ is exactly $l'_v$ can be represented by the forward and backward variables as below, which is also termed as the *occupation probability*:

$$\sum_{\substack{\pi \in \mathcal{B}^{-1}(\mathbf{l}) \\ \pi_t = l'_v}} p(\pi|\mathbf{x}) = \sum_{\substack{\pi \in \mathcal{B}^{-1}(\mathbf{l}) \\ \pi_t = l'_v}} \prod_{t'=1}^{T} y_{\pi_{t'}}^{t'} = \frac{\alpha(t, v) \cdot \beta(t, v)}{y_{l'_v}^t} \tag{4}$$

---

[3]$|\cdot|$ is the length function.

In addition, for any constant $1 \leq t \leq T$, the choices of $\pi_t$ for any possible path $\pi \in \mathcal{B}^{-1}(\mathbf{l})$ are restricted to the elements in $\mathbf{l}'$ and these choices are exclusive[4]. Thus, enumerating all index $v$ with Eq.4 will consider all possible paths and then the training objective of CTC can be written as:

$$P(\mathbf{l}|\mathbf{x}) = \sum_{\pi \in \mathcal{B}^{-1}(\mathbf{l})} p(\pi|\mathbf{x}) = \sum_{v=1}^{2U+1} \sum_{\substack{\pi \in \mathcal{B}^{-1}(\mathbf{l}) \\ \pi_t = \mathbf{l}'_v}} p(\pi|\mathbf{x}) = \sum_{v=1}^{2U+1} \frac{\alpha(t,v) \cdot \beta(t,v)}{y_{l'_v}^t} \tag{5}$$

The computation of the $\alpha(t,v)$ and $\beta(t,v)$ is recursive and the CTC gradients can also be computed in close form using the forward and backward variables. Details are presented in Appendix A.

## 3 BAYES RISK CTC

In this part, a general formulation of the proposed Bayes risk CTC (BRCTC) criterion is presented in section 3.1. Examples about how paths can be grouped to fit the forward-backward process are presented in section 3.2. Finally, we demonstrate how the proposed BRCTC with customizable risk designs can be used to tackle two different practical problems in section 3.3 and section 3.4.

### 3.1 GENERAL FORMULATION

CTC prediction has two functionalities: predicting the target sequence $\mathbf{l}$ and predicting the hard alignment $\text{ali}(\mathbf{l}, \mathbf{x})$ between the input and the target sequences. The former is implemented by discriminating all paths $\pi \in \mathcal{B}^{-1}(\mathbf{l})$ from the sequence set $\mathcal{L}'^T$ since feeding each path into $\mathcal{B}$ will yield the target $\mathbf{l}$. In vanilla CTC, however, no constraint is posed to the latter since Eq.1 treats all paths equally and the choice of the dominant path, a.k.a. the alignment, is left unpredictable. To make the alignment prediction controllable is exactly to break this equality and intentionally select the desired paths among $\mathcal{B}^{-1}(\mathbf{l})$. To this end, a Bayes risk function $r(\pi)$ is adopted to enforce the characteristics of the desired paths. The modified CTC objective can be written as:

$$J_{\text{brctc}}(\mathbf{l}, \mathbf{x}) = \sum_{\pi \in \mathcal{B}^{-1}(\mathbf{l})} [p(\pi|\mathbf{x}) \cdot r(\pi)] \tag{6}$$

The revised objective is termed Bayes risk CTC (BRCTC) due to the adoption of the Bayes risk function. Note when $r(\pi) = 1$, BRCTC becomes equivalent to vanilla CTC. Since BRCTC is still maximizing the posteriors of the paths in $\mathcal{B}^{-1}(\mathbf{l})$, ideally it will preserve the models' target prediction performance.

Directly applying the risk function to each path is still prohibitive as enumerating all possible paths is computationally impractical. To this end, the forward-backward algorithm in the vanilla CTC is inherited to efficiently compute the BRCTC objective. Inheriting the forward-backward process will lead the design of the risk function to a divide-and-conquer paradigm. Under this paradigm, designing the risk function is equivalent to specifying two things: 1) how the paths are divided into groups and 2) what risk values are assigned to each path group. Here the only requirement to achieve compatibility between BRCTC and the forward-backward process is that the summed posterior within any group can be fully represented by the forward-backward variables. Formally, the desired characteristics are about some specific properties of these paths (e.g., the largest index of all non-blank elements within the path). Assume $f(\pi)$ is the concerned property of path $\pi$, then all paths that satisfy $f(\pi) = \tau$ can form a group, where $\tau$ is a possible value of the concerned property. Then, all paths in the same path group will receive the same risk value. Note $r_g(\tau)$ as a function of $\tau$, which is the shared risk value within the group as a replacement of $r(\pi)$. Thus, the BRCTC objective is to enumerate all path groups with the corresponding risk value $r_g(\tau)$ like below. A more detailed explanation is provided in Appendix B.

$$J_{\text{brctc}}(\mathbf{l}, \mathbf{x}) = \sum_{\tau} \sum_{\substack{\pi \in \mathcal{B}^{-1}(\mathbf{l}) \\ f(\pi) = \tau}} [p(\pi|\mathbf{x}) \cdot r(\pi)] = \sum_{\tau} \sum_{\substack{\pi \in \mathcal{B}^{-1}(\mathbf{l}) \\ f(\pi) = \tau}} [p(\pi|\mathbf{x}) \cdot r_g(\tau)] = \sum_{\tau} [r_g(\tau) \cdot \sum_{\substack{\pi \in \mathcal{B}^{-1}(\mathbf{l}) \\ f(\pi) = \tau}} p(\pi|\mathbf{x})]$$

$$\tag{7}$$

### 3.2 EXAMPLES ABOUT HOW PATHS ARE GROUPED

Here we provide two examples of how the paths are grouped to be compatible with the forward-backward process. The first example is in Eq.4, in which paths can be grouped by their choices of $t$-th element $\pi_t$. So the occupation probability is also the summed posterior of a path group, which can be naturally represented by the forward and backward variables.

---

[4]Here we should assume the blank symbols in $\mathbf{l}'$ are different from each other to avoid confusion.

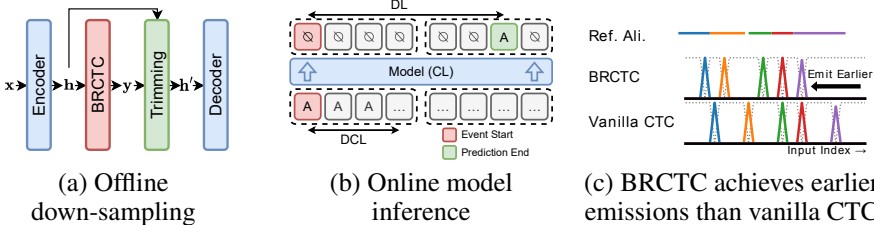

| (a) Offline down-sampling | (b) Online model inference | (c) BRCTC achieves earlier emissions than vanilla CTC |

Figure 2: (a) Down-sampling process using BRCTC criterion. $\mathbf{h}$ is trimmed before being fed into the decoder. (b) Inference process of the online model and its three exclusive sources of latency. DCL: data collecting latency. CL: computational latency. DL: drift latency. (c) Posteriors of online vanilla CTC / BRCTC systems and the reference alignments.

Secondly, a more complicated but useful example is to group the paths according to the ending point of a certain non-blank token. Since any path is an aligning relationship between the input and target sequences, it is common to ask when the prediction of a given non-blank token $l_u = l'_{2u}$ is finished within this path. Formally, with the known constant $u$, the concerned property of $\pi$ can be defined as $\tau = f_u(\pi) = \arg\max_t$ s.t. $\pi_t = l_u = l'_{2u}$ [5]. If so, the summed probability of each path group can be formulated by the forward and backward variables as below. A detailed explanation for this formulation is provided in Appendix C.

$$\sum_{\substack{\pi \in \mathcal{B}^{-1}(\mathbf{l}) \\ f_u(\pi) = \tau}} p(\pi|\mathbf{x}) = \frac{\alpha(\tau, 2u) \cdot \hat{\beta}(\tau, 2u)}{y_{\pi_\tau}^\tau}, \text{ s.t. } \hat{\beta}(\tau, 2u) = \begin{cases} \beta(\tau, 2u) - \beta(\tau+1, 2u) \cdot y_{\pi_\tau}^\tau, \text{ if } \tau < T \\ \beta(\tau, 2u), \hspace{3cm} \text{Otherwise} \end{cases}$$

(8)

Combine Eq.7, 8, for any constant $u$, path groups with different $\tau$ and the corresponding risk values $r_g(\tau)$ are enumerated as below. This strategy of grouping paths is adopted in section 3.3 and 3.4.

$$J_{\text{brctc}}(\mathbf{l}, \mathbf{x}) = \sum_{\tau=1}^{T} r_g(\tau) \cdot \frac{\alpha(\tau, 2u) \cdot \hat{\beta}(\tau, 2u)}{y_{\pi_\tau}^\tau}$$

(9)

### 3.3 APPLICATION: DOWN-SAMPLE

A key problem for a series of offline seq2seq tasks (like ASR, ST) is that the input sequence is much longer than the output sequence, a.k.a., $|\mathbf{x}| \gg |\mathbf{l}|$(Gaido et al., 2021). If an encoder is used to process $\mathbf{x}$ into the encoder hidden output $\mathbf{h}$, this can partially be interpreted as $|\mathbf{h}| \gg |\mathbf{l}|$. For the mainstream encoder-decoder architectures in seq2seq tasks, the inference cost of the decoder is highly correlated with $|\mathbf{h}|$. Thus, the over-length of $\mathbf{h}$ leads to redundancy in the inference computation. The proposed BRCTC is capable of reducing the length of the $\mathbf{h}$ to save inference costs. The workflow is shown in Fig.2.a. The non-blank spikes of CTC posterior $\mathbf{y}$ are pushed to the earlier time-stamps using BRCTC, then the posterior $\mathbf{y}$ is adopted as a reference to trim the $\mathbf{h}$ into the shorter $\mathbf{h}'$.

Pushing all non-blank spikes to the early input units (like in Fig.1.a) requires the predicted alignment to finish its prediction of $l_U$ as early as possible. Consider the latter situation in section 3.2 and set $u = U$, the concerned property $\tau$ is the input index where all non-blank elements have completed in the paths. Subsequently, paths with smaller $\tau$ are more preferred so the risk values should be larger. We adopt $r_g(\tau) = e^{-\lambda \cdot \tau / T}$ as the risk function in this application, where $\lambda$ is an adjustable hyper-parameter called *risk factor*. Finally, the training objective is updated as:

$$J_{\text{brctc}}(\mathbf{l}, \mathbf{x}) = \sum_{\tau=1}^{T} e^{-\lambda \cdot \tau / T} \cdot \frac{\alpha(\tau, 2U) \cdot \hat{\beta}(\tau, 2U)}{y_{\pi_\tau}^\tau}$$

(10)

Provided the continuous and highly confident blank predictions in $\mathbf{y}$ like in Fig.1.c, it is reasonable to assume the corresponding elements in $\mathbf{h}$ contain nearly no useful semantics and can be trimmed[6]. Formally, $\mathbf{h}$ is trimmed to $\mathbf{h}' = [\mathbf{h}_1, ..., \mathbf{h}_{m+D}]$, where $m$ is the maximum value of $t$ s.t. $\forall\, t' > t, \mathbf{y}_\varnothing^{t'} > 99\%$; $D = 5$ is a small integer to keep a safety margin for the trimming. The trimmed hidden output $\mathbf{h}'$ is fed into downstream architectures as a replacement of $\mathbf{h}$.

---

[5]There might be some repetitions in $\mathbf{l}$, but we still consider all tokens in $\mathbf{l}$ are different for simplicity. The correlation between any non-blank $\pi_t$ and $l_u$ is clear so this notation will not lead to confusion.

[6]Usually, transforming $\mathbf{h}$ into $\mathbf{y}$ only adopts a simple linear classifier and the softmax function.

## 3.4 APPLICATION: PERFORMANCE-LATENCY TRADE-OFF

Another problem for online seq2seq systems (e.g., streaming ASR) is the trade-off between the transcription performance and the system latency. For online systems with constrained context, better transcription quality requires more context, which, however, will result in longer latency. Assume the input sequence is fed into the system chunk-by-chunk (Shi et al., 2021), this paper defines the total latency as the sum of three exclusive sources as shown in Fig.2.b: **Data collecting latency (DCL)**: the time to wait before the input signal forms a chunk. This depends on the model design. **Computational latency (CL)**: the time consumed by model inference. Only this latency depends on the hardware performance. **Drift latency (DL)**: the difference between the input indexes when an event starts and when its prediction ends[7]. This is learned during the model training. The formal definition of the latency sources and further explanation are in Appendix F.

The proposed BRCTC can guide the model to emit non-blank spikes at early input indexes (see Fig.2.c) so that the drift latency (DL) can be reduced. As a benefit, it provides a better overall performance-latency trade-off than the vanilla CTC systems (more discussion is in section 4.3). Formally, if a non-blank token $l_u$ is required to be emitted earlier, the concerned property $\tau$ is exactly the ending point of its prediction within the path. Thus, a tailor-made training objective for the token $l_u$ is:

$$J'_{\text{brctc}}(\mathbf{l}, \mathbf{x}, u) = \sum_{\tau=1}^{T} e^{-\lambda \cdot (\tau - \tau')/T} \cdot \frac{\alpha(\tau, 2u) \cdot \hat{\beta}(\tau, 2u)}{y_{\pi_\tau}^\tau}, \text{ s.t. } \tau' = \arg\max_\tau \frac{\alpha(\tau, 2u) \cdot \hat{\beta}(\tau, 2u)}{y_{\pi_\tau}^\tau} \quad (11)$$

The design of the group risk function $r_g(\tau)$ is still the exponential decay function but with an extra bias $\tau'$. Without this bias, the absolute values of $J'_{\text{brctc}}(\mathbf{l}, \mathbf{x}, u)$ will be unbalanced for different token $l_u$.[8] To guide every token $l_u$ to emit earlier requires the considerations of all $u$. So the global training objective to maximize is then transformed into:

$$J_{\text{brctc}}(\mathbf{l}, \mathbf{x}) = \frac{1}{U} \cdot \sum_{u=1}^{U} \log J'_{\text{brctc}}(\mathbf{l}, \mathbf{x}, u) \quad (12)$$

## 4 EXPERIMENTS

Our experiments are mainly designed to examine the two applications of the proposed BRCTC criterion. The experimental setup is introduced in section 4.1. The BRCTC down-sampling method and performance-latency trade-off are validated in section 4.2 and section 4.3 respectively. BRCTC is generalized to MT and ST in section 4.4. Visualization and its analysis are in section 4.5.

### 4.1 EXPERIMENT SETUP

**Datasets:** Experiments are mainly conducted for ASR, but MT and ST are also included. For ASR, the experiments are on Aishell-1 (Bu et al., 2017), Aishell-2 (Du et al., 2018), Wenetspeech (Zhang et al., 2022) and Librispeech (Panayotov et al., 2015). The volumes of these datasets range from 178 hours to 10k hours. Librispeech is in English and the others are in Mandarin. For MT and ST, IWSLT14 (Cettolo et al., 2012) and MuST-C-v2 En-De (Di Gangi et al., 2019) are adopted.

**Models:** For all tasks, Hybrid CTC/Attention model (Watanabe et al., 2017; Yan et al., 2022) is evaluated. For offline ASR, Transducer (Graves, 2012) plus CTC architecture is also evaluated.

**Training and Decoding:** For training, a two-stage method is proposed for the offline down-sampling application and vanilla CTC is directly replaced by BRCTC in the online systems. For offline decoding, default algorithms with decoder recurrence (Watanabe et al., 2017; Graves, 2012; Yan et al., 2022) are used to demonstrate the reduction of inference cost; for online decoding, CTC greedy search is adopted to better measure the transcription performance and latency. Our BRCTC implementation depends on the differentiable finite-state transducer[9] (Hannun et al., 2020).

**Evaluation Metrics:** To compare the transcription performance, CER (for Mandarin) and WER (for English) are reported for ASR task; detokenized case-sensitive BLEU (Post, 2018) is reported for MT and ST tasks. To compare the computational cost of offline models, the real-time factor (RTF), the down-sampling factor (DSF, a.k.a., $|\mathbf{h}'|/|\mathbf{h}|$) and its oracle (a.k.a., $|\mathbf{l}|/|\mathbf{h}|$) are reported. To analyze the latency, time is measured for hardware-independent latency while the RTF is the indicator of hardware latency. Implementation details are in appendix D and E for reproducibility.

---

[7]See Fig.2.c, this latency is only computed with the input unit indexes, not on the real-world timeline.

[8]Usually, the path groups with $\tau$ being equal or close to $\tau'$ make most of the contributions in Eq.11. For different $u$, the $\tau'$, along with the risk values for these major path groups, is different. So different $\tau'$ leads to unbalance in the absolute values of $J'_{\text{brctc}}(\mathbf{l}, \mathbf{x}, u)$. As a remedy, taking $\tau'$ as a bias ensures that the path group with maximum posterior always receives the risk of 1.0 regardless of $u$.

[9]Our implementation is based on K2 toolkit: https://github.com/k2-fsa/k2

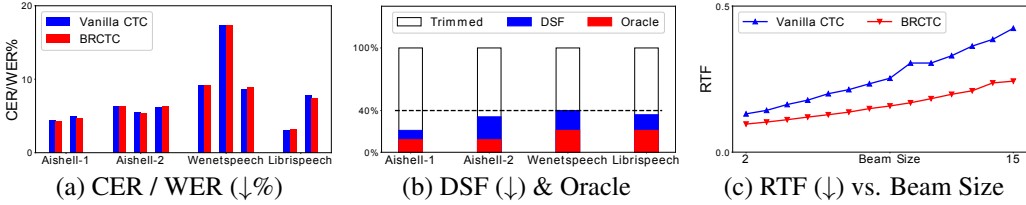

(a) CER / WER ($\downarrow$%)   (b) DSF ($\downarrow$) & Oracle   (c) RTF ($\downarrow$) vs. Beam Size

Figure 3: Transducer performance on various ASR tasks w/o BRCTC down-sampling.

## 4.2 RESULTS ON BRCTC DOWN-SAMPLING FOR OFFLINE ASR SYSTEM

This part evaluates the effectiveness of BRCTC down-sampling method on the offline ASR task. Our main results of the Transducer plus CTC model are plotted in Fig.3. The tendency for the Hybrid CTC/Attention model is similar. The complete results are in Appendix G. Firstly, the transcription performance of the Transducers plus CTC or BRCTC is reported in Fig.3.a. As suggested, with datasets in varying volumes and different languages, adopting BRCTC does not result in noticeable variance in the transcription quality. Secondly, Fig.3.b. demonstrates the effectiveness of the down-sampling process. The encoder hidden output $\mathbf{h}$ is trimmed by at least 60% and the trimmed $\mathbf{h}'$ is roughly as 1.5~2.5 times long as the $\mathbf{l}$ only[10]. So the length mismatch of input and output sequences is significantly alleviated. Thirdly, since the inference cost of the decoder highly depends on the length of $\mathbf{h}$, replacing $\mathbf{h}$ by $\mathbf{h}'$ will reduce the inference cost remarkably. For the label-synchronous decoding of hybrid CTC/Attention (Watanabe et al., 2017), the computational cost of each decoding step is reduced since the expense of attention computation will be smaller. For the frame-synchronous decoding of Transducers (Graves, 2012), the inference cost is reduced since the number of frames in decoding is reduced linearly along with $|\mathbf{h}|$. As the inference cost of the decoder is also sensitive to the beam size, we sweep the beam size from 2 to 15 on Librispeech test-other set. As Fig.3.c. suggests, the relative inference cost reduction becomes larger with the beam size growing.

## 4.3 RESULTS ON BRCTC PERFORMANCE-LATENCY TRADE-OFF FOR ONLINE ASR SYSTEM

This part evaluates the trade-off between the transcription performance and the latency w/o the adoption of BRCTC on streaming ASR task. Our main observation is shown in Fig.4. Fig.4.a reflects the trade-off between the DCL and DL. Firstly, the models designed with smaller DCL will encounter larger DL since the models are not confident with the highly restricted context and will wait for longer input before decisions. This DCL-DL relationship suggests the system with extremely low overall latency is not feasible by only designing low DCL (a.k.a., small chunk size). Secondly, the adoption of BRCTC achieves consistent DL reduction since it tries to push the spikes of all non-blank tokens to be emitted earlier. Thirdly, the DL gap between vanilla CTC and BRCTC increases along with DCL increasing, since a longer chunk provides a higher performance ceiling for BRCTC[11] while the DL for vanilla CTC is always larger than 0ms to explore all accessible contexts. Fig.4.b reflects the trade-off between the DCL and CER. As expected, CTC / BRCTC systems with smaller DCL degrade more in their transcription quality due to the more restricted context. The adoption of BRCTC also results in CER degradation compared with its baseline, since earlier emissions will also restrict the accessible context.

Summarizing Fig.4.a&b leads to Fig.4.c, in which the trade-off between all hardware-independent latency (DCL + DL) and the transcription quality (CER) is demonstrated. The adoption of BRCTC provides several benefits: 1) building the system with extremely low total latency (only 280ms, the pink circle), which is not feasible for vanilla CTC due to the seesaw-like relationship between DCL and DL. 2) Achieving both lower CER and smaller latency than vanilla CTC (the green circles). Specifically, BRCTC provides an alternative solution for online applications: increasing DCL with a larger chunk size and reducing DL using BRCTC to meet the latency budget and achieve a better overall performance-latency trade-off.

The alternative solution of using a larger DCL (larger chunk size) also provides an extra benefit from the perspective of hardware. As suggested by Chen et al. (2021) and also presented in Fig.4.d, a larger chunk size can better explore the advantage of parallel computing and then achieve lower RTF.

---

[10]Note that $\mathbf{x}$ has been sub-sampled by 4 times when being encoded into $\mathbf{h}$(Bérard et al., 2018). Then our BRCTC down-sampling method is conducted on the $\mathbf{h}$. The two down-sampling methods are used together.

[11]The DL can be negative due to the look-ahead mechanism of the model.

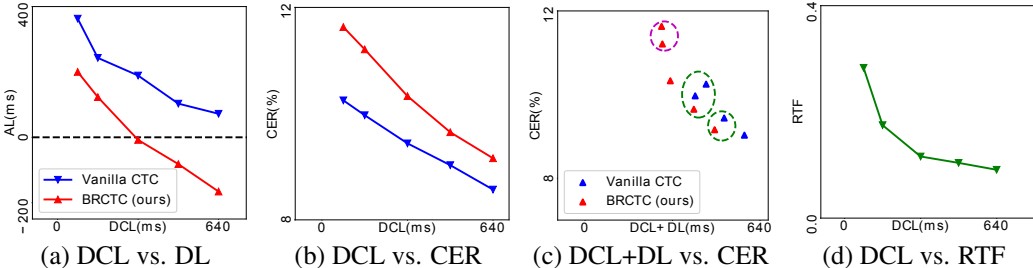

| (a) DCL vs. DL | (b) DCL vs. CER | (c) DCL+DL vs. CER | (d) DCL vs. RTF |

Figure 4: Aishell-2 trade-off between transcription performance and latency w/o BRCTC.

This means the hardware may have more time being idle or switching among multiple processes during serving. Using a larger chunk size will slightly increase CL, but our observation from Fig.4.c still holds. Appendix H provides more detailed results.

## 4.4 GENERALIZING BRCTC TO ST

This part demonstrates that the proposed BRCTC can also be generalized to other seq2seq tasks like ST. The ST results are in table 1. Consistent with our ASR experiments, adopting BRCTC down-sampling in offline ST 1) achieves comparable transcription quality with the vanilla system; 2) reduces the length of $\mathbf{h}$ by 42% and accelerates the inference by 27%. The proposed BRCTC is also generalized to MT task in Appendix I.

| System | Transcription Quality (BLEU↑) | | Down-sampling Effectiveness | |
|---|---|---|---|---|
| | COMMON | HE | DSF(↓) / Oracle | RTF (↓) |
| Attention + Vanilla CTC (Yan et al., 2022) | **29.3** | 28.5 | - | 0.51 |
| Attention + BRCTC (ours) | 29.1 | **28.7** | 0.58 / 0.25 | **0.37** (-27%) |

Table 1: ST performance on MuST-C-V2 English-German dataset w/o BRCTC down-sampling

## 4.5 VISUALIZATION

Following Graves et al. (2006), Fig.5 compares the evolution of the CTC distributions and the their gradient. At the beginning of training, the distribution is roughly unified and the gradient is smooth along the time-axis. Note the gradient for all non-blank tokens is similar only except for the last token (the green line and the red arrow): for the last token, the BRCTC gradient for a larger frame index will be penalized in order to push the last token to the earlier indexes. After two epochs of training, the gradient will localize. Although the models are not sure what the non-blank tokens exactly are, the places where the non-blank tokens will spike are roughly determined. At this stage, the BRCTC model has already known the spikes should all happen at the very left. After convergence, the gradient is close to zero and the CTC distributions become peaky. For BRCTC, the spikes are all concentrated on the left as expected. We also find the down-sampling process is implemented mainly by the last two encoder layers. More visualization is provided in Appendix J.

## 5 DISCUSSION

**Correlation between target prediction and alignment prediction**: This work observes that target predictions in high quality can be obtained from CTC posteriors which do not necessarily encode accurate alignment prediction. Theoretically, as all paths in Eq.1 are treated equally in vanilla CTC and the convergence of vanilla CTC can be achieved with any path being dominant, there are multiple solutions for the alignment prediction sub-task. Additionally, selecting each of these solutions can hardly interfere with the transcription quality, since each of the paths will yield the correct target prediction after the blank and repeat removal. Experimentally, our experiments suggest that vanilla CTC systems with significantly drifted alignment prediction can still preserve the transcription quality (see Appendix K); experimental in section 4.2 further demonstrate that BRCTC achieves competitive transcription results like vanilla CTC with extremely unreasonable alignment prediction.

**Monotonic assumption, MT & ST tasks and rearranging ability**: An underlying assumption of CTC is the monotonic assumption, which requires that, if any $x_t$ and $x_{t'}$ are mapped to two non-blank tokens $l_u$ and $l_{u'}$ respectively with $t < t'$, then there must have $u < u'$. Conventionally, this assumption restricts CTC from being applied to seq2seq tasks whose alignment is not monotonic, like MT and ST. However, this constraint can be softened by deploying self-attention encoder architectures which allow to implicitly reorder the semantics of $\mathbf{h}$ and make it roughly monotonic with respect to $\mathbf{l}$ (Chuang et al., 2021). Besides, our method in section 3.3 can also be viewed as a process to rearrange the semantics of $\mathbf{h}$ even though the relative order of non-blank spikes is kept. To

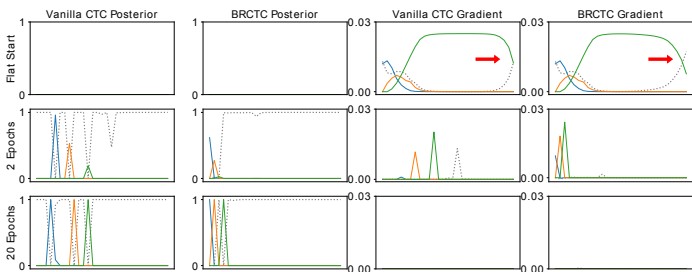

Figure 5: Evolution of CTC distribution **y** and the corresponding gradients on $\log \mathbf{y}$. BRCTC with the down-sampling method is used.

sum up, although this work mainly addresses the CTC training criterion, we believe the rearranging ability of attention-based neural networks is a key factor in BRCTC's functionality.

## 6 RELATED WORKS

The proposed BRCTC is an extension of CTC. There are several existing criteria and frameworks that can be partially viewed as CTC extensions. To alleviate the *conditional independence assumption* of CTC yields Transducer (Graves, 2012), RNA (Sak et al., 2017) and their path-modified extensions (Mahadeokar et al., 2021; Kuang et al., 2022; Shinohara & Watanabe, 2022; Yu et al., 2021; Kim et al., 2021); to equip CTC with discriminative ability yields LF-MMI with CTC topology (Hadian et al., 2018); to exploit the non-auto-regressive nature of CTC yields the non-auto-regressive ASR (Higuchi et al., 2020) and MT (Qian et al., 2021) architectures; to explore various topologies yields multiple CTC variants (Zhao & Bell, 2022; Laptev et al., 2022). To exploit partially labeled and multi-labeled data yields STC (Pratap et al., 2022) and GTC (Moritz et al., 2021). None of the works aforementioned try to control the alignment prediction of CTC. Some pioneer works can be interpreted as preliminary attempts to control CTC alignment prediction. Ghorbani et al. (2018); Kurata & Audhkhasi (2019) achieve aligned CTC posterior spikes among heterogeneous models by learning the spike time-stamps of a teacher model. However, the spikes of the teacher model itself cannot be controlled. Zeyer et al. (2020); Senior et al. (2015); Plantinga & Fosler-Lussier (2019) improve CTC and Transducer models by injecting external alignment supervisions, but obtaining these alignments consumes extra effort. By contrast, the proposed BRCTC is customizable for general purposes in an end-to-end fashion and depends on neither teacher models nor additional information. Previous literature has also discussed the basic properties of CTC from multiple perspectives, like its peaky behavior (Zeyer et al., 2021), alignment drift (Sak et al., 2015) and the properties of blank symbol (Zhao & Bell, 2022; Bluche et al., 2015). To our knowledge, The correlation between the target prediction and alignment prediction has not been seriously discussed before this work.

## 7 LIMITATION

The proposed BRCTC has a known limitation. The gradient of BRCTC is obtained by the naive chain rule (as the forward-backward process only adopts addition and multiplication operations, this is feasible). Thus, the gradient computation needs to trace back the forward-backward process, which then results in an increase in the training cost.

## 8 CONCLUSION

This work is motivated by our experimental observation that a CTC system with drifted alignment prediction still preserves competitive transcription ability, which inspires us the possibility of making CTC alignment prediction controllable to serve the needs in various applications. An extension of CTC called BRCTC is then proposed to select the predicted alignment among all possible paths and the design of the risk function is left customizable to fulfill the task-specific needs of different applications. For the offline model, the adoption of BRCTC leads to inference cost reduction since the length of the intermediate hidden output can be down-sampled. For the online model, BRCTC provides a better performance-latency trade-off. We verify the effectiveness of the proposed BRCTC on multiple sequence-to-sequence tasks with various datasets, languages, and model architectures.

## 9 ACKNOWLEDGEMENT

This work used the Extreme Science and Engineering Discovery Environment (XSEDE) (Towns et al., 2014), which is supported by National Science Foundation grant number ACI-1548562. Specifically, it used the Bridges system (Nystrom et al., 2015), which is supported by NSF award number ACI-1445606, at the Pittsburgh Supercomputing Center (PSC).

## 10 REPRODUCIBILITY STATEMENT

We are taking various measures to ensure the reproducibility of our experiments:

- Code is released as the complementary material of this submission.
- Details of all experiments are clarified in Appendix E.

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

# A APPENDIX: DETAILS OF THE FORWARD-BACKWARD PROCESS AND THE CLOSE-FORM GRADIENT OF VANILLA CTC

This appendix describes the recursive computation of the forward-backward algorithm and the close form gradient of vanilla CTC.

For the forward variable $\alpha(t, v)$, the recursive process is:

$$\alpha(t, v) = \begin{cases} [\alpha(t-1, v) + \alpha(t-1, v-1)] \cdot y_{l'_v}^t, & \text{if } l'_v = \varnothing \text{ or } l'_v = l'_{v-2} \\ [\alpha(t-1, v) + \alpha(t-1, v-1) + \alpha(t-1, v-2)] \cdot y_{l'_v}^t, & \text{Otherwise} \end{cases} \tag{13}$$

with the initial condition:

$$\alpha(1, 1) = y_\varnothing^1; \qquad \alpha(1, 2) = y_{l'_2}^1; \qquad \alpha(1, v) = 0, \quad \forall v > 2 \tag{14}$$

Symmetrically, for the backward variable $\beta(t, v)$, the recursive process is:

$$\beta(t, v) = \begin{cases} [\beta(t+1, v) + \beta(t+1, v+1)] \cdot y_{l'_v}^t, & \text{if } l'_v = \varnothing \text{ or } l'_v = l'_{v+2} \\ [\beta(t+1, v) + \beta(t+1, v+1) + \beta(t+1, v+2)] \cdot y_{l'_v}^t, & \text{Otherwise} \end{cases} \tag{15}$$

with the initial condition:

$$\beta(T, 2U+1) = y_\varnothing^T; \qquad \beta(T, 2U) = y_{l'_{2U}}^T; \qquad \beta(T, v) = 0, \quad \forall v < 2U \tag{16}$$

After all forward and backward variables are computed, the CTC gradient can be computed in close form. Firstly, for the posterior of any path $\pi$, its gradient w.r.t. the $y_k^t$ is:

$$\frac{\partial p(\pi|\mathbf{x})}{\partial y_k^t} = \frac{\partial \prod_{t'=1}^T y_{\pi_{t'}}^{t'}}{\partial y_k^t} = \begin{cases} (\prod_{t'=1}^{t-1} y_{\pi_{t'}}^{t'}) \cdot (\prod_{t'=t+1}^T y_{\pi_{t'}}^{t'}) & \text{if } k = \pi_t \\ 0 & otherwise \end{cases} \tag{17}$$

Then consider the occupation probability in Eq.4 and include the two products above into the forward and backward variables, its gradient w.r.t. the output $y_k^t$ is:

$$\frac{\partial \sum_{\pi \in \mathcal{B}^{-1}(l); \pi_t = l'_v} p(\pi|\mathbf{x})}{\partial y_k^t} = \begin{cases} \dfrac{1}{y_k^{t\,2}} \cdot \alpha(t, v) \cdot \beta(t, v) & \text{if } k = \pi_t \\ 0 & otherwise \end{cases} \tag{18}$$

Finally, enumerate all $v$ like in Eq.5, the gradient of CTC is computed as:

$$\frac{p(l|\mathbf{x})}{\partial y_k^t} = \frac{1}{y_k^{t\,2}} \cdot \sum_{v \in lab(l', k)} \alpha(t, v) \cdot \beta(t, v) \tag{19}$$

where $lab(l', k)$ indicates where the $k$ occurs: $lab(l', k) = \{v : l'_v = k\}$

## B  FURTHER EXPLANATION OF BRCTC GENERAL FORMULATION

This appendix provides a more detailed explanation of the BRCTC general formulation. The equations are provided below.

$$
\begin{aligned}
J_{\text{brctc}}(\mathbf{l}, \mathbf{x}) &= \sum_{\pi \in \mathcal{B}^{-1}(\mathbf{l})} [p(\pi|\mathbf{x}) \cdot r(\pi)] \\
&= \sum_{\tau} \sum_{\substack{\pi \in \mathcal{B}^{-1}(\mathbf{l}) \\ f(\pi)=\tau}} [p(\pi|\mathbf{x}) \cdot r(\pi)] \\
&= \sum_{\tau} \sum_{\substack{\pi \in \mathcal{B}^{-1}(\mathbf{l}) \\ f(\pi)=\tau}} [p(\pi|\mathbf{x}) \cdot r_g(\tau)] \\
&= \sum_{\tau} [r_g(\tau) \cdot \sum_{\substack{\pi \in \mathcal{B}^{-1}(\mathbf{l}) \\ f(\pi)=\tau}} p(\pi|\mathbf{x})]
\end{aligned}
\tag{20}
$$

The first line is the original definition of BRCTC, in which each path $\pi$ along with its risk value $r(\pi)$ is considered and the product is summed for all paths. The second line is to group the paths with identical concerned property $\tau$. In this stage, the risk value for each path is still $r(\pi)$. The third line suggests the risk values for all paths within a group are identical and only depend on the concerned property $\tau$. So the risk value is changed from $r(\pi)$ to $r_g(\tau)$ within the group. The final line is to extract the common factor $r_g(\tau)$ out of the summation within the group. This suggests we can first compute the summed posterior of all paths within the group and then apply the risk value for that group in one go.

**Example:** we further provide a naive example to illustrate how paths are grouped and how the risk values are assigned to each group. Assume the input length is 3 (index from 1 to 3) and the target sequence is [A, B]. Then, all possible paths and their posteriors are listed in the first and the second rows of the table below respectively.

We also assume the concerned property $\tau = f(\pi)$ is the time-stamp when the prediction of token $B$ finishes. Thus, the value of the concerned property for each path is listed in the third row of the table below.

Paths with the identical $\tau$ value should form a group. So the path $AB\varnothing$ forms a group with a single element while the remained paths form another group. For paths within a group, the risk value should be identical. The risk value for each group $r_g(\tau)$ is user-defined. We simply set $r_g(2) = 1.0$ and $r_g(3) = 0.8$. Then the risk for each path $r(\pi)$ is then set in the fourth row of the table below.

| Path($\pi$) | AB$\varnothing$ | ABB | AAB | A$\varnothing$B | $\varnothing$AB |
|---|---|---|---|---|---|
| $p(\pi|\mathbf{x})$ | 0.3 | 0.1 | 0.2 | 0.0 | 0.1 |
| $\tau$ | 2 | 3 | 3 | 3 | 3 |
| $r(\pi)$ or $r_g(\tau)$ | 1.0 | 0.8 | 0.8 | 0.8 | 0.8 |

Finally, formulating all these processes in equations will be:

$$
\begin{aligned}
\mathbf{J}_{\text{brctc}} &= p(AB\varnothing) \cdot r(AB\varnothing) + p(ABB) \cdot r(ABB) + p(AAB) \cdot r(AAB) \\
&\quad + p(A\varnothing B) \cdot r(A\varnothing B) + p(\varnothing AB) \cdot r(\varnothing AB) \quad \textit{\#Initial formulation} \\
&= [p(ABB) \cdot r(ABB) + p(AAB) \cdot r(AAB) + p(A\varnothing B) \cdot r(A\varnothing B) + p(\varnothing AB) \cdot r(\varnothing AB)] \\
&\quad + [p(AB\varnothing) \cdot r(AB\varnothing)] \quad \textit{\#grouping} \\
&= [p(ABB) \cdot r_g(3) + p(AAB) \cdot r_g(3) + p(A\varnothing B) \cdot r_g(3) + p(\varnothing AB) \cdot r_g(3)] \\
&\quad + [p(AB\varnothing) \cdot r_g(2)] \quad \textit{\#replace } r(\pi) \textit{ by } r_g(\tau) \\
&= [p(ABB) + p(AAB) + p(A\varnothing B) + p(\varnothing AB)] \cdot r_g(3) \\
&\quad + [p(AB\varnothing)] \cdot r_g(2) \quad \text{\#extract common factors} \\
&= (0.1 + 0.2 + 0.0 + 0.1) \cdot 0.8 + 0.3 \cdot 1.0 = 0.62
\end{aligned}
\tag{21}
$$

# C   APPENDIX: THE EXPLANATION FOR GROUPING STRATEGY FORMULATION

This appendix explains the formulation in section 3.2. The property function is set to $\tau = f_u(\pi) = \arg\max_t$ s.t. $\pi_t = l_u = l'_{2u}$.

Firstly, for the scenario $\tau < T$, the property function can be translated into the condition $\pi_\tau = l'_{2u}$ and $\pi_{\tau+1} \neq l'_{2u}$. Thus, the summed probability of the path group is formulated as:

$$\sum_{\substack{\pi \in \mathcal{B}^{-1}(1) \\ f_u(\pi)=\tau}} p(\pi|\mathbf{x}) = \sum_{\substack{\pi \in \mathcal{B}^{-1}(1) \\ \pi_\tau = l'_{2u}; \pi_{\tau+1} \neq l'_{2u}}} p(\pi|\mathbf{x}) = \sum_{\substack{\pi \in \mathcal{B}^{-1}(1) \\ \pi_\tau = l'_{2u}; \pi_{\tau+1} \neq l'_{2u}}} \prod_{t'=1}^{T} y_{\pi_{t'}}^{t'}$$

$$= \sum_{\substack{\pi \in \mathcal{B}^{-1}(1) \\ \pi_\tau = l'_{2u}}} \prod_{t'=1}^{T} y_{\pi_{t'}}^{t'} - \sum_{\substack{\pi \in \mathcal{B}^{-1}(1) \\ \pi_\tau = l'_{2u}; \pi_{\tau+1} = l'_{2u}}} \prod_{t'=1}^{T} y_{\pi_{t'}}^{t'}$$

$$= \frac{\alpha(\tau, 2u) \cdot \beta(\tau, 2u)}{y_{\pi_\tau}^\tau} - \alpha(\tau, 2u) \cdot \beta(\tau+1, 2u) \tag{22}$$

Set:

$$\hat{\beta}(\tau, 2u) = \beta(\tau, 2u) - \beta(\tau+1, 2u) \cdot y_{\pi_\tau}^\tau \tag{23}$$

The summed probability then becomes:

$$\sum_{\substack{\pi \in \mathcal{B}^{-1}(1) \\ f_u(\pi)=\tau}} p(\pi|\mathbf{x}) = \frac{\alpha(\tau, 2u) \cdot \hat{\beta}(\tau, 2u)}{y_{\pi_\tau}^\tau} \tag{24}$$

Secondly, $\tau = T$ suggests that:

$$\sum_{\substack{\pi \in \mathcal{B}^{-1}(1) \\ f_u(\pi)=\tau}} p(\pi|\mathbf{x}) = \alpha(\tau, 2u) = \frac{\alpha(\tau, 2u) \cdot \beta(\tau, 2u)}{y_{\pi_\tau}^\tau} \tag{25}$$

To sum up, the summed probability of the path group with the given $\tau$ and $u$ is

$$\sum_{\substack{\pi \in \mathcal{B}^{-1}(1) \\ f_u(\pi)=\tau}} p(\pi|\mathbf{x}) = \frac{\alpha(\tau, 2u) \cdot \hat{\beta}(\tau, 2u)}{y_{\pi_\tau}^\tau} \tag{26}$$

with

$$\hat{\beta}(\tau, 2u) = \begin{Bmatrix} \beta(\tau, 2u) - \beta(\tau+1, 2u) \cdot y_{\pi_\tau}^\tau, & \text{if } \tau < T \\ \beta(\tau, 2u), & \text{Otherwise} \end{Bmatrix} \tag{27}$$

# D APPENDIX: THE TWO-STAGE METHOD FOR BRCTC DOWN-SAMPLING

Figure 6: An illustration of the proposed two-stage method for BRCTC down-sampling.

This appendix describes the proposed two-stage method for BRCTC down-sampling. Details are described in Fig.6. The first stage mainly follows the architecture in Watanabe et al. (2017): the encoder-decoder model is jointly optimized by both CTC loss and cross-entropy loss. The decoder part is adopted in order to be reused in the second stage. Intermediate CTC technique (Lee & Watanabe, 2021) is consistently adopted so that the whole encoder is split into two parts with an identical number of layers, and the hidden outputs from both the intermediate layer (a.k.a., $\mathbf{h_1}$) and the final layer (a.k.a., $\mathbf{h_2}$) are supervised by the CTC criterion. In this stage, all CTC criteria are our proposed BRCTC. In the second stage, the weights of the two encoder parts and the CTC classifier are preserved, but several things are changed. First, the $\mathbf{h_1}$ is trimmed into $\mathbf{h_1'}$ using the CTC posterior $\mathbf{y_1}$ as the reference, so the length of the intermediate hidden output is reduced. Second, the CTC loss for $\mathbf{h_2}$ is the vanilla CTC loss since $\mathbf{h_2}$ does not need to be trimmed again. Thirdly, the $\mathbf{h_2}$ is then fed into downstream modules and losses.

The design of downstream modules and losses depends on the specific tasks and system architectures. For ASR and MT tasks with the Hybrid CTC/Attention architecture, the attention decoder and the cross-entropy loss are adopted and the weight can be inherited from the first stage. For ASR with the Transducer architecture, the prediction network and the joint network are randomly initialized and the Transducer loss is adopted. For the ST task with the Hybrid CTC/Attention architecture, the translation encoder and the attention decoder are randomly initialized and two extra losses are adopted: the vanilla CTC loss and the cross-entropy loss. Note in ST tasks, the text labels from both the source language and the target language are provided. CTC loss for Encoder 1 and Encoder 2 still adopts the source language labels like in the first stage, but the translation encoder and the attention decoder are supervised by the target language labels ($\mathbf{tgt\_l}$). For all architectures, the global training objective is the weighted sum of all loss items.

Note that, in the second stage, it is possible to use vanilla CTC or BRCTC consistently. However, when vanilla CTC is consistently adopted, the down-sampling factor will degrade gradually along with the second-stage training; when BRCTC is consistently adopted, a slight performance degradation is observed. All hyper-parameters of this two-stage method are presented in Appendix E.

# E  APPENDIX: REPRODUCIBILITY

This appendix describes the details of all experiments for reproducibility.

- **Datasets**: all statistics of the datasets are in table 2.
- **Models and Features**: all model architectures and the input features are presented in table 3, table 4, table 6 and table 5.
- **Training and BRCTC settings**: Adam (Kingma & Ba, 2014) optimizer with the learning rate being inverse square root decaying (Vaswani et al., 2017) is adopted. Details for the optimization and the setting for BRCTC are shown in table 7. All experiments are conducted on Nvidia V100 GPUs.
- **Decoding**: For the online ASR model, we adopt greedy CTC decoding. For all other offline models, the decoding configurations are in table 8. All the inference jobs are conducted on Intel(R) Xeon(R) Platinum 8255C CPU (2.5GHz). The RTFs are calculated by the first 1/88 data in the corresponding test sets. External language models in any form are not used in inference. Checkpoints from the last 10 epochs (for ASR) or the 10 epochs with the best validation accuracy (for ST) are averaged for evaluation.

| Dataset | Task | #Hours | #Pairs | Language | Units |
|---|---|---|---|---|---|
| Aishell-1 (Bu et al., 2017) | ASR | 178 | 120k | Mandarin | 4231 Char. |
| Aishell-2 (Du et al., 2018) | ASR | 1k | 962k | Mandarin | 5214 Char. |
| Wenetspeech (Zhang et al., 2022) | ASR | 10k | 14M | Mandarin | 6267 Char. |
| Librispeech (Panayotov et al., 2015) | ASR | 960 | 281k | English | 500 BPE |
| IWSLT De-En (Cettolo et al., 2012) | MT | - | 160k | German-English | 10k BPE (shared) |
| IWSLT Es-En (Cettolo et al., 2012) | MT | - | 160k | Spanish-English | 10k BPE (shared) |
| MuST-C-V2 En-De (Di Gangi et al., 2019) | ST | 450 | 250k | English-German | 500 BPE / 4k BPE |

Table 2: Dataset description of ASR / MT / ST tasks.

| Acoustic input features: FBank + Pitch | | | |
|---|---|---|---|
| Frame_length | 25ms | Frame_shift | 10ms |
| Fbank_dim | 80 | Pitch_dim | 3 |
| Text input: Embedding | | | |
| Embedding_size | 512 | | |
| Acoustic data augmentation: speed perturbation and specaugment(Park et al., 2019) | | | |
| Num time masks | 2 | Time mask length | 40 |
| Num frequency masks | 2 | Frequency mask length | 30 |
| Max time warp | 5 | speed perturbation factors | [0.9, 1.0, 1.1] |
| Encoder: Conformer (Gulati et al., 2020): | | | |
| Num layer | 12 | Num attention head | 4 |
| Attention dim | 512 | Feed-forward dim | 2048 |
| Num CNN module kernel | 31 | CNN down sample | 4x |
| Decoder for hybrid CTC/Attention architecture: Transformer (Vaswani et al., 2017): | | | |
| Num layer | 6 | Num attention head | 4 |
| Attention dim | 512 | Feed-forward dim | 2048 |
| Decoder for Transducer (Graves, 2012) plus CTC architecture: | | | |
| Prediction network | LSTM | LSTM hidden size | 512 |
| LSTM num layer | 1 | Joint network | Linear |
| Joint network dim | 512 | | |
| Hybrid CTC/attention (Watanabe et al., 2017) architecture: | | | |
| CTC loss weight | 0.3 | Attention loss weight | 0.7 |
| Attention label smooth | 0.1 | | |
| Transducer (Graves, 2012) + CTC architecture: | | | |
| CTC loss weight | 0.5 | Transducer loss weight | 1.0 |
| Others: | | | |
| Inter. CTC (Lee & Watanabe, 2021) | 6-th | Inter. CTC weight | 0.3 |
| Dropout rate | 0.1 | | |

Table 3: Offline ASR system configuration

| Acoustic input features: FBank + Pitch | | | |
|---|---|---|---|
| Frame_length | 25ms | Frame_shift | 10ms |
| Fbank_dim | 80 | Pitch_dim | 3 |
| Text input: Embedding | | | |
| Embedding_size | 512 | | |
| Acoustic data augmentation: speed perturbation and specaugment(Park et al., 2019) | | | |
| Num time masks | 2 | Time mask length | 40 |
| Num frequency masks | 2 | Frequency mask length | 30 |
| Max time warp | 5 | speed perturbation factors | [0.9, 1.0, 1.1] |
| Encoder: Emformer (Shi et al., 2021): | | | |
| Num layer | 12 | Num attention head | 4 |
| Attention dim | 512 | Feed-forward dim | 2048 |
| Memory bank length | 4 | Left context | 320ms |
| Chunk size: Right Context | 2:1 or 1:1 | Chunk size | [80, 160, 320, 480, 640]ms |
| Decoder for hybrid CTC/attention architecture: Transformer (Vaswani et al., 2017): | | | |
| Num layer | 6 | Num attention head | 4 |
| Attention dim | 512 | Feed-forward dim | 2048 |
| Hybrid CTC/Attention (Watanabe et al., 2017) architecture: | | | |
| CTC loss weight | 0.3 | Attention loss weight | 0.7 |
| Attention label smooth | 0.1 | | |
| Others: | | | |
| Dropout rate | 0.1 | | |

Table 4: Online ASR system configuration

| Acoustic input features: FBank | | | |
|---|---|---|---|
| Frame_length | 25ms | Frame_shift | 10ms |
| Fbank_dim | 80 | | |
| Text input: Embedding | | | |
| Embedding_size | 512 | | |
| Acoustic data augmentation: speed perturbation and specaugment(Park et al., 2019) | | | |
| Num time masks | 5 | Time mask length | 5% of T |
| Num frequency masks | 2 | Frequency mask length | 27 |
| Max time warp | 5 | speed perturbation factors | [0.9, 1.0, 1.1] |
| Encoder: Conformer (Gulati et al., 2020): | | | |
| Num layer | 12 | Num attention head | 4 |
| Attention dim | 256 | Feed-forward dim | 2048 |
| Num CNN module kernel | 31 | CNN down sample | 4x |
| Translation Encoder: Conformer (Gulati et al., 2020): | | | |
| Num layer | 6 | Num attention head | 4 |
| Attention dim | 512 | Feed-forward dim | 2048 |
| Num CNN module kernel | | | |
| Decoder for hybrid CTC/attention architecture: Transformer (Vaswani et al., 2017): | | | |
| Num layer | 6 | Num attention head | 4 |
| Attention dim | 512 | Feed-forward dim | 2048 |
| Hybrid CTC/attention (Watanabe et al., 2017) architecture: | | | |
| ASR CTC loss weight | 0.3 | ST CTC loss weight | 0.21 |
| ST Attention loss weight | 0.49 | Attention label smooth | 0.1 |
| Others: | | | |
| Inter. CTC (Lee & Watanabe, 2021) | 6-th | Inter. CTC weight | 0.3 |
| Dropout rate | 0.1 | | |

Table 5: ST system configuration

| Text input: Embedding | | | |
|---|---|---|---|
| Embedding_size | 512 | | |
| MT encoder: LegoNN Encoder (Dalmia et al., 2022): | | | |
| Num layer before up-sampling | 6 | Num layer after up-sampling | 6 |
| Attention dim | 512 | Feed-forward dim | 1024 |
| Num attention head | 4 | Up-sampling rate | 3 |
| Decoder for hybrid CTC/attention architecture: Transformer (Vaswani et al., 2017): | | | |
| Num layer | 6 | Num attention head | 4 |
| Attention dim | 512 | Feed-forward dim | 1024 |
| Hybrid CTC/attention (Watanabe et al., 2017) architecture: | | | |
| CTC loss weight | 0.3 | Attention loss weight | 0.7 |
| Attention label smooth | 0.1 | | |
| Others: | | | |
| Inter. CTC (Lee & Watanabe, 2021) | 6-th | Inter. CTC weight | 0.3 |
| Dropout rate | 0.3 | | |

Table 6: MT system configuration

| Dataset | $\lambda$ (offline) | $\lambda$ (online) | #epochs | peak lr | #warmup iter. | #GPU | Max Global Batch size |
|---|---|---|---|---|---|---|---|
| Aishell-1 | 10 | 20 | 50+50 | 3e-4 | 25k | 8 | 400 seconds |
| Aishell-2 | 10 | 20 | 50+50 | 3e-4 | 25k | 8 | 400 seconds |
| Wenetspeech | 30 | - | 50+50 | 3e-4 | 25k | 32 | 3200 seconds |
| Librispeech | 100 | - | 50+50 | 3e-4 | 25k | 8 | 400 seconds |
| IWSLT | 50 | - | 150+50 | 2e-3 | 10k | 4 | 8M bin |
| MuST-C-V2 | 50 | - | 60+40 | 1e-3 | 25k | 8 | 6M bin |

Table 7: Optimization strategy and BRCTC settings. Epochs 50+50 means 50 epochs for both the first and the second stages. If there is only one stage (baseline offline systems, online systems), the number of epochs is the sum. Bin for each example: bin = input length · output length · input dimension.

| Dataset | Beam Size | CTC weight | Attention weight | Length reward |
|---|---|---|---|---|
| Aishell-1 | 10 | 0.5 | 0.5 | 0 |
| Aishell-2 | 10 | 0.5 | 0.5 | 0 |
| Wenetspeech | 10 | 0.5 | 0.5 | 0 |
| Librispeech | 10 | 0.5 | 0.5 | 0 |
| IWSLT | 5 | {0, 0.2, 0.4, 0.6, 0.8} | 1 - CTC weight | {0, 0.2, 0.4, 0.6, 0.8} |
| MuST-C-V2 | 10 | {0, 0.2, 0.4, 0.6, 0.8} | 1 - CTC weight | {0, 0.2, 0.4, 0.6, 0.8} |

Table 8: Offline decoding configurations. Numbers in curly braces {} indicate grid search. For grid search we report the best results.

# F APPENDIX: FORMAL DEFINITION OF THE LATENCY SOURCES AND FURTHER EXPLANATION

This appendix provides the formal definitions for the three latency sources in section 3.4. All kinds of latency are computed as the expected value and are at the token-level. Note we are using the Emformer (Shi et al., 2021), which defines three parts of context: *left_context*, *chunk* and *right_context*. The left context is the observed history and is irrelevant to the latency. The chunk is the current context. The adoption of the right context allows a look-ahead mechanism.

- Data Collecting Latency (DCL): DCL = chunk_size$/2$ + right_context, which is the mean time to wait before the collected frames can form a chunk.

- Computational Latency (CL): CL = chunk_size $*$ RTF, which represents the rough inference time for a chunk.

- Drift Latency (DL): DL = $\tau - \hat{\tau}$, where $\tau$ is the ending unit index of the token prediction in the path; $\hat{\tau}$ is the starting unit index of that token obtained by DNN-HMM systems. To ensure every token has its reference, we only consider the tokens in the *longest common subsequence* between the reference transcription and predicted hypothesis. The DL can be negative due to the look-ahead mechanism.

We further provide an example to explain that the three latency sources are exclusive and should be accumulated. Assume an event happens at the step with the input index of $\hat{\tau}$. The input unit for the $\hat{\tau}$ step will not enter the model for inference until the following units are collected enough to form a chunk, which results in the DCL. After the chunk forms, the computation on the model will take some time, which is the CL. Even though the inference process of the $\hat{\tau}$ step has been finished, the output posterior $\mathbf{y}^{\hat{\tau}}$ usually will not predict the event due to the index drift shown in Fig.2.c. Instead, the model may predict the event at another input index $\tau$. So the Event prediction cannot be emitted until the inference process of the $\tau$ unit has been finished, which is also a latency. The gap between $\tau$ and $\hat{\tau}$ is the DL. Note the DL only depends on the input index, not the real-world timeline.

We would also like to note that this decomposition of latency sources ideally ignores the other sources like communication, feature extraction, model loading, etc., as these latency sources are usually marginal or out of the scope of seq2seq tasks.

## G APPENDIX: DETAILED EXPERIMENTAL RESULTS ON OFFLINE DOWN-SAMPLING

| | Aishell-1 | Aishell-2 | Wenetspeech | Librispeech |
|---|---|---|---|---|
| | 178h, Mandarin | 1kh, Mandarin | 10kh, Mandarin | 960h, English |
| System | dev / test | android / ios / mic | dev / meeting / net | t-clean / t-other |
| Attention + CTC | **4.26 / 4.74** | 6.33 / 5.48 / 6.29 | **9.44 / 15.97** / 9.11 | 3.02 / **7.72** |
| Attention + BRCTC (ours) | 4.30 / 4.75 | **6.13 / 5.34 / 6.03** | 9.59 / 16.86 / **9.04** | 3.15 / **7.63** |
| Transducer + CTC | 4.47 / 4.92 | 6.39 / 5.47 / **6.18** | **9.20 / 17.34 / 8.61** | **3.05** / 7.79 |
| Transduder + BRCTC (ours) | **4.33 / 4.73** | **6.36 / 5.35** / 6.30 | **9.20** / 17.41 / 8.87 | 3.14 / **7.41** |

Table 9: ASR results on the models' transcription performance w/o BRCTC down-sampling method. All models adopt auto-regressive decoding algorithms. CER/WER (↓) are reported. The beam size is set to 10 consistently.

| | Aishell-1 | | Aishell-2 | | Wenetspeech | | Librispeech | |
|---|---|---|---|---|---|---|---|---|
| | test | | test-ios | | test-net | | t-other | |
| System | RTF | DSF / Oracle | RTF | DSF / Oracle | RTF | DSF / Oracle | RTF | DSF / Oracle |
| Attention + CTC | 1.19 | - | 1.07 | - | 1.91 | - | 1.97 | - |
| Attention + BRCTC (ours) | **0.94** | 0.21 / 0.12 | **0.95** | 0.34 / 0.12 | **1.53** | 0.37 / 0.21 | **1.38** | 0.35 / 0.21 |
| Transducer + CTC | 0.37 | - | 0.42 | - | 0.48 | - | 0.31 | - |
| Transducer + BRCTC (ours) | **0.21** | 0.20 / 0.12 | **0.22** | 0.29 / 0.12 | **0.33** | 0.40 / 0.21 | **0.18** | 0.36 / 0.21 |

Table 10: Evaluations results on the models' inference cost w/o BRCTC down-sampling method. The real-time factor (RTF ↓), the down-sampling factor (DSF ↓) and its oracle are reported. The beam size is set to 10 consistently. The maximum inference cost reduction happens in Aishell-2 Transducer case, in which the RTF is reduced from 0.42 to 0.22 (47% relative reduction).

## H APPENDIX: DETAILED EXPERIMENTAL RESULTS ON ONLINE PERFORMANCE-LATENCY TRADE-OFF

| λ | DCL+DL+CL(ms) | Hardware-Independent | | | Hardware-Dependent | | CER% | Marker |
|---|---|---|---|---|---|---|---|---|
| | | DCL (ms) | DL (ms) | DCL+DL (ms) | RTF | CL (ms) | Greedy Search | |
| | Aishell-1 test | | | | | | | |
| | 474 | 240 | 206 | 446 | 0.176 | 28 | 6.88 | ⋆ |
| | 480 | 120 | 336 | 456 | 0.305 | 24 | 7.19 | ⋆ |
| 0 | 614 | 480 | 94 | 574 | 0.128 | 40 | 6.28 | ⋆⋆ |
| | 850 | 720 | 80 | 800 | 0.106 | 50 | 5.77 | |
| | 1090 | 960 | 72 | 1032 | 0.092 | 58 | 5.55 | |
| | 315 | 240 | 47 | 287 | 0.176 | 28 | 8.10 | ▲ |
| | 339 | 120 | 195 | 315 | 0.305 | 24 | 7.97 | ▲ |
| 20 | 440 | 480 | -80 | 400 | 0.128 | 40 | 7.23 | ▲ |
| | 501 | 960 | -517 | 443 | 0.092 | 58 | 6.40 | ⋆ |
| | 570 | 720 | -200 | 520 | 0.106 | 50 | 6.31 | ⋆⋆ |
| λ | Aishell-2 test-android | | | | | | | |
| | 431 | 160 | 243 | 403 | 0.175 | 28 | 9.97 | ⋆ |
| | 465 | 80 | 363 | 443 | 0.283 | 22 | 10.25 | ⋆ |
| 0 | 546 | 320 | 189 | 509 | 0.116 | 37 | 9.44 | ⋆⋆ |
| | 632 | 480 | 103 | 583 | 0.104 | 49 | 9.03 | |
| | 770 | 640 | 72 | 712 | 0.091 | 58 | 8.57 | |
| | 302 | 80 | 200 | 280 | 0.283 | 22 | 11.63 | ▲ |
| | 311 | 160 | 123 | 283 | 0.175 | 28 | 11.21 | ▲ |
| 20 | 349 | 320 | -8 | 312 | 0.116 | 37 | 10.33 | ▲ |
| | 447 | 480 | -82 | 398 | 0.104 | 49 | 9.65 | ⋆ |
| | 532 | 640 | -166 | 474 | 0.091 | 58 | 9.16 | ⋆⋆ |

Table 11: Trade-off between the transcription performance and the latency for online CTC models. $\lambda$: risk factor of BRCTC; DCL: data collecting latency; CL: computational latency; DL: drift latency; CER: character error rate; RTF: real-time factor. Latency data is computed for greedy search only. ⋆ and ⋆⋆ represent cases for comparison. ▲ represents the extremely low latency cases that cannot be achieved by vanilla CTC. The minimum overall latency achieved by BRCTC is only 302ms (Aishell-2). Compared with its vanilla baseline whose minimum overall latency is 431ms, the BRCTC achieves a 30% overall latency reduction relatively.

## I    APPENDIX: DETAILED EXPERIMENTAL RESULTS ON MT TASKS

This appendix presents the experimental results on MT tasks. As suggested in table 12, BRCTC can reduce the length of $\mathbf{h}$ to 63% and save 27% inference cost. At the same time, the model with BRCTC still preserves the competitive transcription ability (very close BLEU scores).

However, several things are noticeable. For MT tasks, the lengths of $\mathbf{x}$ and $\mathbf{l}$ are usually close. Commonly, there is no need to conduct the down-sampling on $\mathbf{h}$ in MT tasks. This experiment follows the setting in (Yan et al., 2022), where CTC is integrated as an auxiliary criterion. To ensure $|\mathbf{h}| \geq |\mathbf{l}|$ in CTC computation, the $\mathbf{x}$ are up-sampled for 3 times when being encoded into $\mathbf{h}$. In this case, the down-sampling process will be needed.

| System | Transcription Quality (BLEU↑) | | Down-sampling Effectiveness | |
|---|---|---|---|---|
| | De-En | Es-En | DFS(↓) / Oracle | Rel. Inference Time (↓) |
| Attention + Vanilla CTC (Yan et al., 2022) | **31.9** | 37.9 | - | 1.00 |
| Attention + BRCTC (ours) | 31.7 | **38.0** | 0.63 / 0.34 | 0.73 |

Table 12: MT performance on IWSLT14 dataset w/o BRCTC down-sampling.

## J    APPENDIX: MORE VISUALIZATION

This appendix provides more visualization results on 1) the gradient analysis of BRCTC in online applications and 2) the attention analysis of BRCTC down-sampling process.

Fig.7 compares the evolution of the CTC distributions and their gradients on the online performance-latency trade-off application. Most of the observations are similar to those in Fig.5 except 1) at the beginning of training, the gradients for each non-blank token, rather than the last non-blank token only, are interfered with by the adoption of Bayes risk function. The gradient peaks of BRCTC are shifted to the left. 2) after 2 epochs of training, the gradients will also be localized, but BRCTC has not learned the places where the emission will happen. After convergence, the emissions of BRCTC will be earlier. We add the input indexed to show its difference from vanilla CTC.

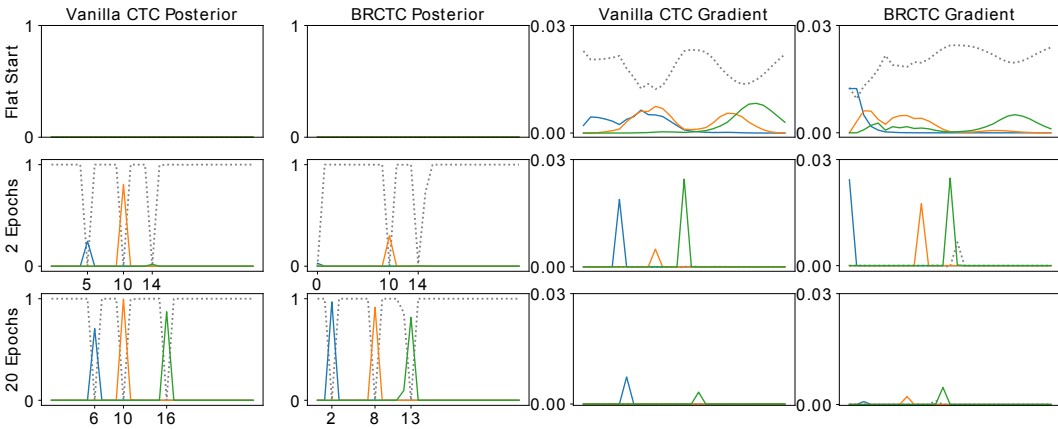

Figure 7: Evolution of CTC distribution $\mathbf{y}$ and the corresponding gradients on $\log \mathbf{y}$. BRCTC with emission latency alleviation method is used.

Fig.8 demonstrates the encoder self-attention weights from different layers and attention heads. As shown in the figure, the down-sampling process is completed mainly in the last two layers before the BRCTC criterion (the 5-th and 6-th layers). The attention weights shown in the red boxes suggest how the semantics of input units with large input indexes are aggregated to the output units with small indexes. Observations like this are mainly in the last two layers before BRCTC so we assume the global context is fully explored in other layers.

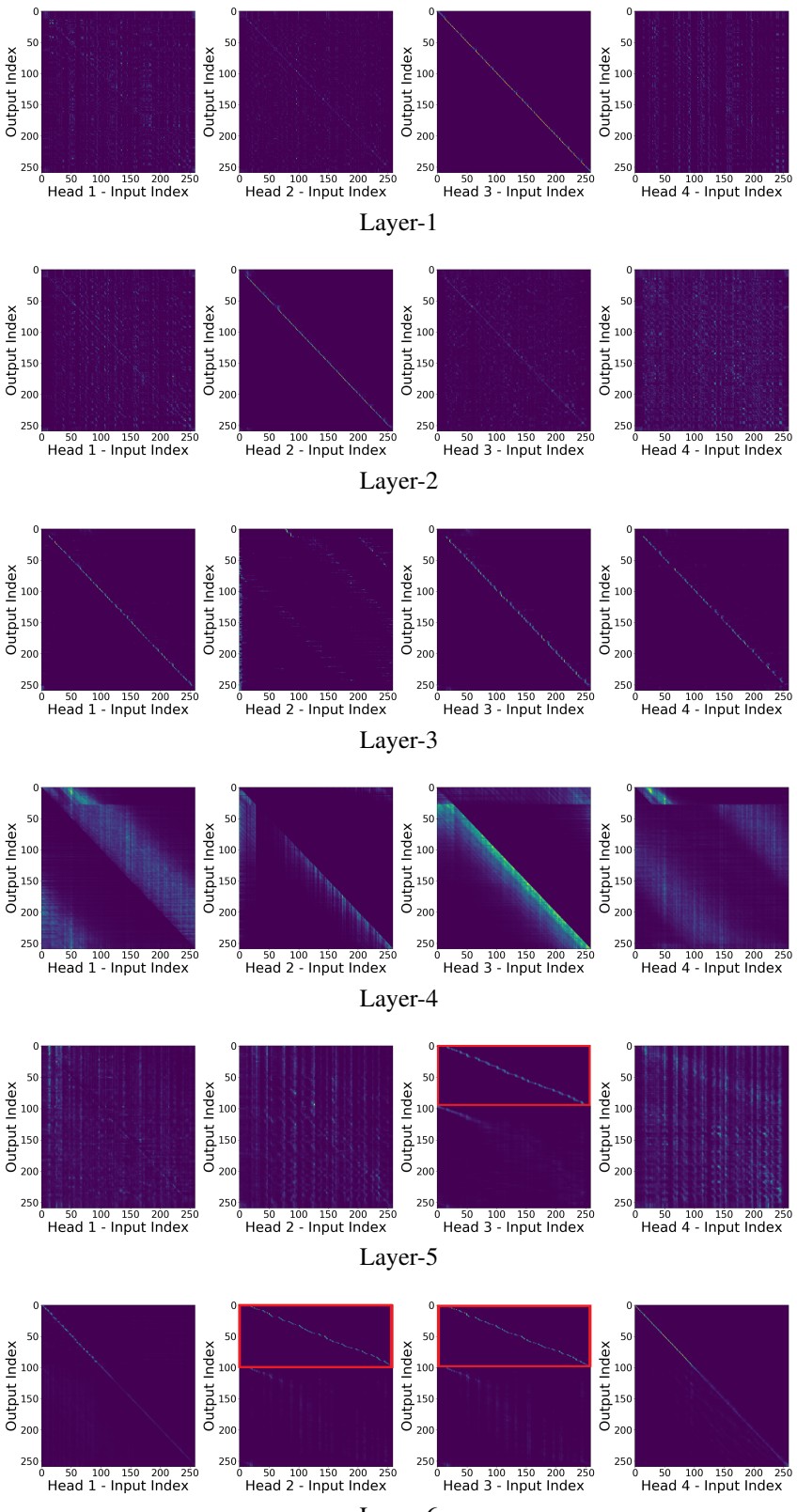

Figure 8: Encoder attention plots from different layers and attention heads. Data from Librispeech test-clean set, utterance 1089-134686-0000. Down-sampling is conducted in the last two layers before BRCTC only (see the red boxes).

# K    APPENDIX: ALIGNMENT DRIFT OF VANILLA CTC SYSTEM

This appendix demonstrates that systems even trained with **vanilla CTC** are predicting the alignment that drifts significantly. We draw several CTC posteriors and their reference alignment obtained by DNN-HMM systems (the colored bars) in Fig.9. All figures are obtained from Aishell-2 test-android set.

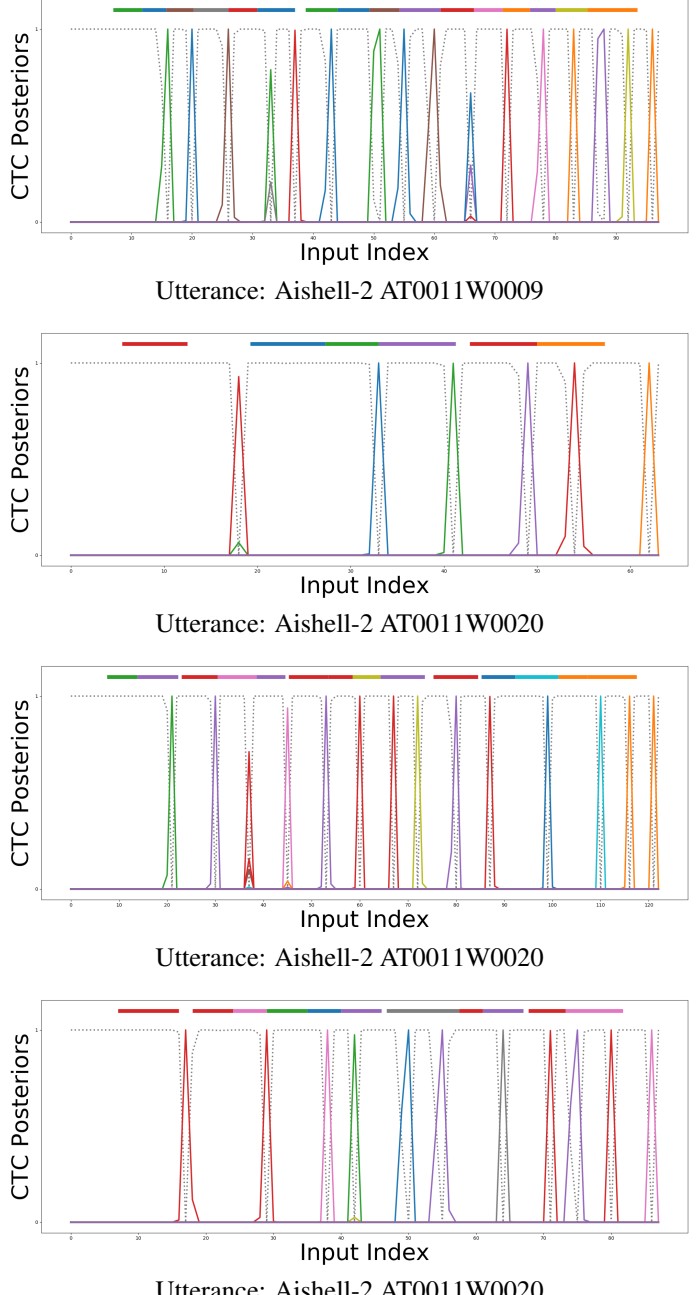

Utterance: Aishell-2 AT0011W0009

Utterance: Aishell-2 AT0011W0020

Utterance: Aishell-2 AT0011W0020

Utterance: Aishell-2 AT0011W0020

Figure 9: CTC posteriors and their reference alignment predicted by **vanilla CTC** systems. Colored bars are the reference alignments obtained by DNN-HMM systems. The predicted alignment drift significantly compared with the DNN-HMM reference alignment.

