# OpenReview forum: "BAYES RISK CTC: CONTROLLABLE CTC ALIGNMENT IN SEQUENCE-TO-SEQUENCE TASKS"
_ICLR.cc/2023/Conference — ICLR 2023 poster_

### Official Review · Reviewer_GviL · 2022-10-23

**Confidence:** 3
**Correctness:** 4
**Technical Novelty And Significance:** 3
**Empirical Novelty And Significance:** 3
**Recommendation:** 8

**Clarity, Quality, Novelty And Reproducibility:**

The paper is clearly written, high quality, novel and the authors have released their code for reproducibility.


**Strength And Weaknesses:**

The paper is well written and the proposed techniques are very well formulated mathematically. The authors provide abundant materials in the appendix and also release their code. CTC alignment is an important problem for ASR and may have consequences on a variety of downstream applications. The authors investigate two of them and the results are impressive. The proposed technique itself is quite elegant and mathematically grounded. I appreciate the effort put in by the authors to build the mathematical framework and providing all the details of the experiments in the appendix.

Comments:
1. There shouldn’t be a ‘t’ in the subscript for ‘p’ in equation (2)
2. I may be missing something here, but shouldn’t the final result in equation (5) be a product over all ‘t’s.
3. The arg max operation defined in the first paragraph of page 5 is not very clear. There is a boolean operation inside the argmax(.) where elements will be either 0 or 1, what will argmax return in such a case?
4. Typo: page 6 last line, DFS -> DSF

Questions:
1. Why did the authors not compare against some simple baselines:

    a) Downsampling: Using a CNN for subsampling a speech sequence such that the new sequence length is similar to the one obtained by the controlled alignment. The CNN is like a feature extractor from speech. Fox example, wav2vec2.0 uses 20ms windows, what happens if we use a >20ms window and lower the sequence length? How much performance is degraded and how does it compare with the proposed model?

    b) Online Latency reduction: Plantinga et al. [1] have a simple solution for emitting phones sooner where they have an “alignment loss which encourages outputs only when features do not resemble silence.” It would also be a good idea to cite the above paper.

2. Can such an alignment controlling mechanism be used to train RNN transducers which are also very popular and use a quite similar algorithm to compute the loss?

3. What are some other applications apart from the two explored where controlling CTC alignment will be useful?

Reference:
[1] Plantinga et al. Towards Real-Time Mispronunciation Detection in Kids' Speech. ASRU 2019.

**Summary Of The Paper:**

This paper re-formulates the widely used CTC criterion for ASR such that the alignments learnt are controllable. This is done by forcing a preference of desirable alignment paths in the CTC loss function. Furthermore, a bayesian risk factor controls this preference. To this end, the forward and backward variables are chosen such that the above preference is satisfied. The authors report the effectiveness of the proposed technique on two tasks: 1) downsampling of speech frames for offline ASR such that inference time is reduced without performance degradation; 2) speeding up online ASR by having earlier emissions without loss in performance. Results on speech translation have also been reported.

**Summary Of The Review:**

This is a good paper which provides a mathematically sound technique to control CTC alignments. This is backed by impressive results on two tasks. The paper may benefit by running some simple baselines.

---

> ### Author Response · Authors · 2022-11-18
> **Response to Reviewer GviL**
>
> **Q1:** There shouldn't be a 't' in the subscript for 'p' in equation
> (2)
>
> **A:** Thanks for pointing out this. This typo is fixed
>
> **Q2:** I may be missing something here, but shouldn't the final result
> in equation (5) be a product over all 't's.
>
> **A:** Thanks for asking this. We believe that summing over all $t$ in
> equation 5, with a fixed $v$, is not valid. An implicit requirement of
> this summing process is that each path should be considered once and
> only once. Firstly, with a fixed $v$, $\pi_{t} = {l'}_v $ does not exclude
> the possibility of ${{\pi}\_{t+1}} = {l'}\_{v}$, which means some paths are
> counted more than once when we are enumerating all $t$. Secondly, when
> ${l'}_v$ is a blank symbol, it is not guaranteed that ${l'}_v$ will be
> included in every path, which means some paths are not counted.
>
> **Q3:** The arg max operation defined in the first paragraph of page 5
> is not very clear. There is a boolean operation inside the argmax(.)
> where elements will be either 0 or 1, what will argmax return in such a
> case?
>
> **A:** We thank this comment. It has been replaced by
> $\tau=f_u(\mathbf{\pi})=\arg\max_{t} \text{ s.t. } \pi_t=l_u$ to exclude
> this ambiguity.
>
> **Q4:** Typo: page 6 last line, DFS $\rightarrow$ DSF
>
> **A:** Thanks for reminding us of this. The typo has been fixed.
>
> **Q5:** Downsampling: Using a CNN for subsampling a speech sequence such
> that the new sequence length is similar to the one obtained by the
> controlled alignment. The CNN is like a feature extractor from speech.
> Fox example, wav2vec2.0 uses 20ms windows, what happens if we use a
> $>$ 20ms window and lower the sequence length? How much
> performance is degraded and how does it compare with the proposed model?
>
> **A:** Thanks for this question. To explore the effect of a larger CNN
> down-sampling factor (DSF), the CNN down-sampling factor is changed from
> 4 (as in our original setup) to 8 (a large number that is rarely used in
> practice). The experiment is conducted on Aishell-1 dataset and the
> results are presented in table 1. As suggested in the table: 1) the
> adoption of larger CNN down-sampling factor results in a slight
> performance degradation; 2) the adoption of the BRCTC down-sampling
> method can effectively reduce the encoder output length regardless of
> the CNN down-sampling factors.
>
> Hint: we still implement the CNN down-sampling using 2D-CNN over the
> Fbank features. We have not implemented the down-sampling by applying
> 1D-CNN over the waveform (like in wav2vec 2.0) in order to keep
> consistency with our initial setup.
>
> Table 1: Comparison of different CNN down-sampling factors
> |CNN stride|CNN DSF|Time per frame in **h**|BRCTC down-sampling|len(**h**)|len(**h'**)|Test CER
> |:-:|:-:|:-:|:-:|:-:|:-:|:-:|
> |{2,2}|4|40ms|False|124.1|-|4.74|
> |{2,2}|4|40ms|True|124.1|29.2|4.75|
> |{2,2,2}|8|80ms|False|61.3|-|4.79|
> |{2,2,2}|8|80ms|True|61.3|24.2|4.78|
>
> **Q6:** Online Latency reduction: Plantinga et al. \[1\] have a simple
> solution for emitting phones sooner where they have an "alignment loss
> which encourages outputs only when features do not resemble silence." It
> would also be a good idea to cite the above paper.
>
> **A:** We thank the reviewer for providing us with this paper. We agree
> this paper is related and have added it into the *related works*
> section.
>
> **Q7:** Can such an alignment controlling mechanism be used to train RNN
> transducers which are also very popular and use a quite similar
> algorithm to compute the loss?
>
> **A:** We thank the reviewer for providing us with this additional
> inspiration. Indeed, it is possible to generalize the proposed method to
> the neural transducer since both CTC and transducer are maximizing the
> summed posterior of all valid paths. We prefer to address this problem
> in our future work.
>
> **Q8:** What are some other applications apart from the two explored
> where controlling CTC alignment will be useful?
>
> **A:** We thank the reviewer for encouraging us to explore more possible
> applications of BRCTC.
>
> Although in sections 3.3 and 3.4 we only consider some heuristically
> designed group risk values $r_g(\tau)$, it would be possible to design
> these values by encoding some external supervision information. An
> example is to encode the reference alignment: by adopting the occupation
> condition (like equation 4) as the grouping strategy and adopting group
> risk values that encode the external reference alignment, we attempt to
> obtain accuracy in both recognition results and alignment prediction.
> Our preliminary attempts suggest this direction might be possible: the adoption of BRCTC with a proper risk
> design can remarkably alleviate the disagreement between the predicted
> alignment and its reference. **Please check the PDF version of our response to see the example.**
>
> We would also like to remind the reviewer that the BRCTC is proposed as
> a general framework and the risk design is left customizable. Thus, we encourage our users to design
> the exact Bayes risk function from their own orientations and
> task-specific needs.

---

> > ### Comment · Reviewer_GviL · 2022-11-22
> > **Final comments**
> >
> > I thank the authors for their response and clarifications. I believe this is a strong paper and merits acceptance.

---

### Official Review · Reviewer_NBYg · 2022-10-24

**Confidence:** 4
**Correctness:** 4
**Technical Novelty And Significance:** 3
**Empirical Novelty And Significance:** 2
**Recommendation:** 6

**Clarity, Quality, Novelty And Reproducibility:**

Clarity: good
Quality: good
Novelty: medium
Reproducibility: good

**Strength And Weaknesses:**

Strength
- The method is novel and well-motivated.
- As far as I can see, the method is technically sound and well-justified by the experiments.

Weakness
- The work does not specify a general condition that the grouping function f(\phi) has to meet to enable the forward-backward algorithm to be applied. Instead, it gives an example f(\phi) which is the largest index of non-blank elements in the path. If it is the only case to which the proposed method can be applied, the main value of the work is to speed up CTC inference. Although accelerating CTC inference is also important, the writing of the paper gives readers the impression that it is a general method to control CTC alignment.

**Summary Of The Paper:**

CTC is a fundamental method for many sequence-to-sequence tasks. It works by maximizing the summed probabilities of paths that can be mapped to the target sequence. As CTC treats all paths equally, it cannot generate accurate alignments between its predictions and input sequences. This work proposes a Bayes Risk CTC method (BRCTC) to make CTC alignment controllable by first dividing paths into groups and then specifying preference weights for groups. For offline ASR, BRCTC was shown to effectively reduce the inference cost by down-sampling the hidden vectors. For streaming ASR, BRCTC can accelerate the recognition by forcing the model to emit non-blank spikes as early as possible.



**Summary Of The Review:**

Making CTC output controllable alignment prediction is an interesting topic. The proposed method is novel and technically sound. My main concern is whether the proposed method is applicable to other useful grouping functions except the example given in Sec. 3.2 and if there is a general rule for making a legal grouping function. However, the paper does not include any discussion on it.

---

> ### Author Response · Authors · 2022-11-18
> **Response to Reviwer NBYg**
>
> **A:** We thank the reviewer for acknowledging our contributions. We
> also thank the reviewer for helping us clarify our work. Below we
> explain why the proposed BRCTC is exactly a general method and has the
> potential to be widely used, besides acceleration.
> Firstly, we discuss the compatibility between the forward-backward
> algorithm and our BRCTC. Secondly, we explain why the BRCTC is a general
> and flexible method by providing multiple grouping strategies and risk strategies. We finally show another potential application of BRCTC besides acceleration.
>
> **Compatibility between the forward-backward algorithm and BRCTC:**
> We agree with the reviewer that we should clarify when the BRCTC is
> compatible with the forward-backward algorithm. To our knowledge, the
> only condition to make the BRCTC compatible with the forward-backward
> algorithm is about the grouping strategy: the summed posterior of all
> paths within any path group should be computationally feasible using the
> forward-backward variables, i.e.,
> $\sum_{\mathbf{\pi}\in \mathcal{B}^{-1}(\mathbf{l}), f(\mathbf{\pi})=\tau} p(\mathbf{\pi}|\mathbf{x})$
> can be represented by combining $\alpha$ and $\beta$. Although vanilla
> CTC only considers a special case (equation 4), the BRCTC formulation
> can be generalized to other grouping strategies, like equation 8. Owning
> to the differentiability of $\alpha$ and $\beta$ w.r.t the CTC
> posteriors, the grouping strategies can be designed in a more flexible
> way without worrying about the gradient computation. For better
> clarification, the following sentence is added in section 3.1: *Here the
> only requirement to achieve compatibility between BRCTC and the
> forward-backward process is that the summed posterior within any group
> can be fully represented by the forward-backward variables.*
>
> **Flexibility from grouping strategy design:** We agree with the
> reviewer that we should provide more than one grouping strategy examples
> to claim the generalization. Actually, in section 3.2, we provide two
> grouping strategy examples: 1) considering the state occupation, like in
> equation 4; 2) considering the ending point of a given non-blank token,
> like in equation 8. The latter has been applied to sections 3.3 and 3.4
> and justified by experiments. Although the former is less emphasized in
> our paper, one of its potential applications will be discussed in the
> last second paragraph (other potential applications) of this response.
> Besides the two strategies above, the proposed BRCTC does have other
> possibilities in grouping strategy design. A simple example is that,
> with a fixed $t$, all paths can be divided into two groups concerning
> whether $\pi_t$ is blank or not.
> Theoretically, this grouping strategy has the potential to encourage or
> discourage the blank emission at frame $t$. We have revised section 3.2
> to clarify there are two grouping strategy examples.
>
> **Flexibility from risk strategy design:** We would like to note that
> the flexibility of BRCTC is not only on the grouping strategy design
> $f(\pi)$ but also on the group risk value design $r_g(\tau)$. Even with
> an identical grouping strategy, the BRCTC can serve different purposes
> by changing the group risk value designs. The example is that, although
> the grouping strategy in equation 8 is consistently adopted in both
> sections 3.3 and 3.4, different goals (down-sampling and latency
> reduction) are achieved due to the different group risk value designs.
>
> **Other potential applications:** The paragraphs above have discussed
> the generalization and flexibility of BRCTC from the perspectives of
> both grouping strategy design and risk strategy design. We agree with
> the reviewer that discussing more potential directions for BRCTC
> applications will be beneficial. Thus, we provide another example of
> BRCTC's potential applications: it has the potential to calibrate the
> CTC alignment prediction. Although in sections 3.3 and 3.4 we only
> design the $r_g(\tau)$ without any external information, this design can
> be more flexible if the external information (e.g., reference alignment)
> is provided. Specifically, we may calibrate the CTC alignment by 1)
> adopting the grouping strategy concerning the state occupation, like in
> equation 4, and 2) encoding the reference alignment information into the
> $r_g(\pi)$. With this setting, our preliminary experiments suggest that,
> compared with vanilla CTC, the predicted alignment can achieve better
> agreement with its DNN-HMM reference (**Please check the PDF version of our response to see the example**).
>
> **Conclusion:** Concerning the statements above, we are attempting to convince the reviewer that the
> proposed BRCTC is a general alignment control method rather than a
> special case study. It is obvious that the design of BRCTC is not
> limited to what we mentioned in our paper and in this response, but we
> regret that we cannot enumerate all of these potential possibilities.
> Instead, we would like to encourage our readers to design their
> strategies for their own orientations.

---

### Official Review · Reviewer_TGbL · 2022-10-31

**Confidence:** 4
**Correctness:** 3
**Technical Novelty And Significance:** 3
**Empirical Novelty And Significance:** 3
**Recommendation:** 8

**Clarity, Quality, Novelty And Reproducibility:**

Clarity: could be better.
Quality: good
Reproducibility: code is provided. Hope it will be made public.

**Strength And Weaknesses:**

Strength:
1. Novelty; There are previous studies on different CTC variants but the perspective explored in this paper has not been seen in literature. The idea to
2. Of clear practical benefits; Though most of commercial ASR systems are based on Transducer or LAS, CTC is still widely used as a standalone ASR system or integrated into Transducer/LAS training to provide auxiliary functions. And other fields like MT/ST are also relying on CTC in some scenarios. The reviewer believe the work in this study will result in more flexible CTC based designs.
3. Convincing results; The authors did intensive experiments and show both numeric results and visualization to validate the effectiveness of the proposed BRCTC.

Weakness
1. For online experiments, I think CTC-attention training is unnecessary. It's better to show pure CTC training results.
2. Some are some variants of transducer like (alignment-restricted transducer, prunned Transducer) are also based on modification of forward-backward algorithm to achieve specific properties (latency, computational cost). It's better to cite these papers.
3. For the limitation discussion, the author mentioned the increasing training cost of BRCTC but didn't provide much details.

**Summary Of The Paper:**

This paper introduces a new variant of CTC that can encode conditions to achieve different preferable properties. The authors gave two examples, one is to force the model to emit the last non-blank token as early as possible; the other is to avoid the drift latency of non-blank tokens to provide better trade-off between latency and transcription quality. Both the two examples have practical merit for ASR systems. The authors did intensive experiments on both streaming and non-streaming ASR, ST and MT to show the control ability of the proposed BRCTC. The results are convincing given the experimental settings.

**Summary Of The Review:**

Overall, the reviewer think this is a very good extension to vanilla CTC and can  benefit multiple seq2seq tasks.

---

> ### Author Response · Authors · 2022-11-18
> **Respone to Reviewer TGbL**
>
> **Q1:** For online experiments, I think CTC-attention training is unnecessary. It's better to show pure CTC training results.
>
> **A:** We thank this comment. As instructed, we have conducted similar online experiments as in section 4.4 with the attention decoder removed. The results are reported in table 1 below. Our observations on this group of experiments are consistent with what is claimed in our paper: 1) with a similar latency budget, the BRCTC online method achieves better performance-latency trade-off; 2) the BRCTC achieves the very low overall latency that cannot be achieved by vanilla CTC.
>
> Table 1: Trade-off between the transcription performance and the latency for online CTC models. Experiments are conducted on Aishell-1 dataset. $\lambda$: the risk factor of BRCTC; DCL: data collecting latency; CL: computational latency; DL: drift latency; CER: character error rate; RTF: real-time factor. $\star$ and $\star\star$ represent cases for comparison. $\bullet$ represents the extremely low latency cases that cannot be achieved by vanilla CTC.
> | BRCTC - λ | DCL+DL+CL(ms) | DCL(ms) | DL(ms) | DCL+DL(ms) | RTF  | CL (ms) | Test CER (Greedy Search) | Marker |
> | :---: | :------: | :---: | :----: | :--------: | :--: | :-----: | :-----: | :----: |
> | 0 | 614 | 80  | 513 |593| 0.263| 21 |8.14 | $\star\star$
> | 0 | 529 | 160 | 343 |503| 0.163| 26 |7.24 | $\star$
> | 0 | 512 | 320 | 157 |477| 0.112| 35 |6.88 | $\star$
> | 0 | 613 | 480 | 86  |566| 0.099| 47 |6.43 | $\star\star$
> | 0 | 771 | 640 | 76  |716| 0.086| 55 |6.18 |
> | 10| 293 | 80  | 192 |272| 0.263| 21 |9.98 | $\bullet$
> | 10| 339 | 160 | 153 |313| 0.163| 26 |8.32 | $\bullet$
> | 10| 347 | 320 | -8  |312| 0.112| 35 |8.14 | $\bullet$
> | 10| 517 | 480 | -10 |470| 0.099| 47 |6.88 | $\star$
> | 10| 630 | 640 | -65 |575| 0.086| 55 |6.18 | $\star\star$
>
> **Q2:** Some are some variants of transducer like (alignment-restricted transducer, prunned Transducer) are also based on modification of forward-backward algorithm to achieve specific properties (latency, computational cost). It's better to cite these papers.
>
> **A:** We thank the reviewer for providing us with these papers and agree they are related. The papers below are cited in the *related works* section.
>
>
> * - J. Mahadeokar et al., "Alignment Restricted Streaming Recurrent Neural Network Transducer," 2021 IEEE Spoken Language Technology Workshop (SLT), 2021, pp. 52-59, doi: 10.1109/SLT48900.2021.9383606.*
> * - Kuang, F., Guo, L., Kang, W., Lin, L., Luo, M., Yao, Z., Povey, D. (2022) Pruned RNN-T for fast, memory-eﬀicient ASR training. Proc. Interspeech 2022, 2068-2072, doi: 10.21437/Interspeech.2022-10340*
> * - Shinohara, Y., Watanabe, S. (2022) Minimum latency training of sequence transducers for streaming end-to-end speech recognition. Proc. Interspeech 2022, 2098-2102, doi: 10.21437/Interspeech.2022-10989*
> * - J. Yu et al., "FastEmit: Low-Latency Streaming ASR with Sequence-Level Emission Regularization," ICASSP 2021 - 2021 IEEE International Conference on Acoustics, Speech and Signal Processing (ICASSP), 2021, pp. 6004-6008, \\doi: 10.1109/ICASSP39728.2021.9413803.*
> * - Kim, J., Lu, H., Tripathi, A., Zhang, Q., Sak, H. (2021) Reducing Streaming ASR Model Delay with Self Alignment. Proc. Interspeech 2021, 3440-3444, doi: 10.21437, Interspeech.2021-322*
>
> **Q3:** For the limitation discussion, the author mentioned the increasing training cost of BRCTC but didn't provide much details.
>
> **A:** We thank this comment. With the identical settings in Appendix E, we compare the training cost of Hybrid CTC/Attention systems on Aishell-1 dataset. The results are reported in table 2. The adoption of BRCTC results in 3.9\% computational overhead.
>
>
> Table 2: Training cost comparison between vanilla CTC and BRCTC on Aishell-1 dataset.
> | | Attention + Vanilla CTC | Attention +  BRCTC |
> | :-: | :-: | :-: |
> | Second / Epoch | 883 | 918 (+3.9%)

---

### Official Review · Reviewer_YKWK · 2022-11-03

**Confidence:** 5
**Correctness:** 3
**Technical Novelty And Significance:** 4
**Empirical Novelty And Significance:** 3
**Recommendation:** 8

**Clarity, Quality, Novelty And Reproducibility:**

The paper proposes a novel technique for controlling specific characteristics of CTC alignments. The experiments are conducted on popular public datasets, and all relevant hyper-parameters are listed in the appendix. A reference implementation of the BRCTC loss function is also included, which makes this work reproducible, for example, with the ESPnet toolkit.

The paper is relatively clearly written but contains many minor issues discussed below. Two problems in terms of clarity are:

- Abstract and introduction state that "BRCTC achieves up to 47% inference cost reduction for offline systems without degradation in transcription performance." This statement on its own is misleading. Just changing the loss function does not lead to any speed-up. The speed-up is achieved by architectural changes enabled by BRCTC. Please clarify this in the paper.

- The use of $2u$ in eq. 8 is unintuitive at first read and needs to be explained in the main body of the paper. Please explain that $2u$ is used because $l_u = l'_{2u}$ as you do in Appendix C.

Minor comments:

- The authors cite everything with \citet, but most of the citations are parenthetical and should have been made with \citep.

- $ali(l, x)$ in Sections 2.1 and 3.1 use different formatting conventions.

- There is an extra period after footnote 1 in the abstract. Also use consistent placing of footnotes.

- Figure and section references are inconsistent (Fig 1.a, Fig. 1.b, Fig.3.c, section 4.3, sec.4.2).

- In the sentence "it is common to ask when the prediction of each token $l_u$ is finished within this path." I would recommend replacing "each token $l_u$" with "a given token $l_u$".

**Strength And Weaknesses:**

Strengths:

The paper proposes a novel technique for controlling specific characteristics of CTC alignments using Bayes Risk. Furthermore, the paper shows how to derive the Bayes Risk CTC loss function from the CTC loss function. The conducted experiments show that the proposed method can be used to speed up both offline and online inference.

Weaknesses:

The method is compared only against vanilla CTC. However, since the proposed method effectively reduces the length of the encoder output sequence for offline recognition, it would be good to compare it with naive uniform subsampling of encoder outputs with subsampling factors 2 and 3 to achieve the same down-sampling factor. Furthermore, in the case of online recognition, the paper should mention CTC delay constraints from [1], which uses reference alignments obtained with an external DNN-HMM model. Comparing the BRCTC method with the delay constraints method would be nice. This experiment can be implemented with GTN, which the authors used to implement BRCTC, and delay constraint FSA similar to Kaldi TimeEnforcerFst.

[1] Senior, Andrew, et al. "Acoustic modelling with CD-CTC-sMBR LSTM RNNs." 2015 IEEE Workshop on Automatic Speech Recognition and Understanding (ASRU). IEEE, 2015.

**Summary Of The Paper:**

This paper proposes to use Bayes Risk factors to control specific characteristics of CTC alignments. Specifically, the proposed method encourages the model to move the non-blank predictions towards the beginning of the output sequence. This shift of non-blank predictions enables speed-ups both in offline and online inference. In offline inference, since the non-blank predictions occur at the beginning of the output sequence, the paper proposes to trim all blank predictions after the last non-blank prediction. As a result, the length of decoder input is shorter by 60%, making the decoding process 47% relative faster. In online inference, the speed-up is also achieved by encouraging the model to produce non-blank predictions sooner, thus reducing the drift latency caused by delayed non-blank predictions.

**Summary Of The Review:**

This paper proposes an interesting method which could be very useful for improving the inference speed of ASR models. Therefore, after including the naive down-sampling baseline experiment and fixing the writing/formatting issues mentioned above, I am inclined to accept this paper.

---

> ### Author Response · Authors · 2022-11-18
> **Response to YKWK**
>
> **Q1:** The method is compared only against vanilla CTC. However, since the proposed method effectively reduces the length of the encoder output sequence for offline recognition, it would be good to compare it with naive uniform subsampling of encoder outputs with subsampling factors 2 and 3 to achieve the same down-sampling factor.
>
> **A:** Thanks for this question. As instructed, we implement the uniform down-sampling method over the encoder output. Experiments are conducted on Aishell-1 dataset following our hybrid CTC/attention setup in Appendix E. The results are reported in table 1. As suggested in the table, the proposed BRCTC down-sampling method outperforms the uniform down-sampling method on both DSF and CER consistently.
>
> Table1: Comparison between uniform down-sampling method and the proposed BRCTC. DSF: down-sampling factor, a.k.a., $|\mathbf{h'}| / |\mathbf{h}|$.
>
> | Exp. | DSF | Test CER |
> | :---- | :----: | :----: |
> |Vanilla CTC | - | 4.74 |
> | Vanilla CTC + uniform down-sampling (factor=2) | 0.50 | 4.82 |
> | Vanilla CTC + uniform down-sampling (factor=3) | 0.33 | 4.89 |
> | BRCTC down-sampling (risk factor = 10) | 0.21 | 4.75 |
>
>
>
> **Q2:** Furthermore, in the case of online recognition, the paper should mention CTC delay constraints from [1], which uses reference alignments obtained with an external DNN-HMM model. Comparing the BRCTC method with the delay constraints method would be nice. This experiment can be implemented with GTN, which the authors used to implement BRCTC, and delay constraint FSA similar to Kaldi TimeEnforcerFst.
>
> **A:** We thank the reviewer for providing us with this pioneering approach. This paper has been cited and mentioned in the *related works* section.
> As instructed, we have reproduced the delay constraint approach and the results are reported in table 2. Our implementation is also provided in the latest complementary material (our code). The experiments are still conducted on Aishell-1 dataset. The overall latency (DCL + DL + CL) is restricted to roughly below 500 ms.
>
> As shown in that table, we suppose the recognition performance of delay constraint CTC and the proposed BRCTC are comparable given the similar overall latency budget (see the markers). Note the delay constraint CTC method requires reference alignment but our BRCTC does not, which should be considered a strength of BRCTC.
>
> Table2: Comparison between BRCTC (ours) and delay constraint CTC. All experiments are conducted on Aishell-1 dataset and statistics are based on the test set. DCL: data collecting latency; CL: computational latency; DL: drift latency; CER: character error rate; RTF: real-time factor. $\star$ and $\star\star$ represent cases for comparison.
> | BRCTC - λ | Delay Constraint(ms) | DCL+DL+CL(ms) | DCL(ms) | DL(ms) | DCL+DL(ms) | RTF  | CL (ms) | Test CER (Greedy Search) | Marker |
> | :---: | :----:| :------: | :---: | :----: | :--------: | :--: | :-----: | :-----: | :----: |
> | - | 160| 422 | 120 | 278 |398| 0.305| 24 |7.49 | $\star$
> | - | 240| 441 | 120 | 297 |417| 0.305| 24 |7.41 | $\star$
> | - | 320| 485 | 120 | 341 |461| 0.305| 24 |7.42 | $\star$
> | - | 160| 479 | 240 | 211 |451| 0.176| 28 |7.06 | $\star\star$
> | - | 240| 459 | 240 | 191 |431| 0.176| 28 |7.03 | $\star\star$
> | - | 320| 468 | 240 | 200 |440| 0.176| 28 |7.17 | $\star\star$
> | 20| -  | 315 | 240 | 47  |287| 0.176| 28 |8.10
> | 20| -  | 339 | 120 | 195 |315| 0.305| 24 |7.97
> | 20| -  | 440 | 480 | -80 |400| 0.128| 40 |7.23 | $\star$
> | 20| -  | 501 | 960 | -517|443| 0.092| 58 |6.40 | $\star\star$
> | 20| -  | 570 | 720 | -200|520| 0.106| 50 |6.31
>
> **Q3:** Abstract and introduction state that "BRCTC achieves up to 47\% inference cost reduction for offline systems without degradation in transcription performance." This statement on its own is misleading. Just changing the loss function does not lead to any speed-up. The speed-up is achieved by architectural changes enabled by BRCTC. Please clarify this in the paper.
>
> **A:** We thank the reviewer for helping us to improve the writing. We suppose the speed-up mainly results from the trimming process. Thus, the revisions are listed below.
>
> In abstract: *Experimentally, the proposed BRCTC, along with a trimming approach, enables us to reduce the inference cost of offline models by up to 47\% without performance degradation.*
>
> In introduction: *Experimentally, BRCTC can cooperate with a trimming approach to achieve up to 47\% inference cost reduction for offline systems without degradation in transcription performance.*
>
> **Q4:** The use of $2u$ in eq. 8 is unintuitive at first read and needs to be explained in the main body of the paper. Please explain that $2u$ is used because
> $l_u = {l'}_{2u}$ as you do in Appendix C.
>
> **A:** We thank this suggestion to help our writing be clearer. We have added the notation $l_u = {l'}_{2u}$ in the main text.
>
> **Minor Comments:** We thank the reviewer for checking our paper so carefully. All mentioned minor problems have been fixed accordingly.

---

> > ### Comment · Reviewer_YKWK · 2022-11-28
> > **Final comments**
> >
> > Thank you for addressing my comments and running the additional experiments. I am happy to recommend accepting this paper.

---

### Author Response · Authors · 2022-11-18
**General Response to Reviewers**

We thank the chair for organizing the peer review stage. We also thank all reviewers for acknowledging our contributions and providing us with valuable suggestions. These suggestions are constructive and do help us to improve. Below we address the concerns of each reviewer in a one-by-one style.

Updates:
1) The manuscript and the implementation of BRCTC have been updated according to the reviewers' comments
2) Our response letter is also included in the complementary material. Due to the limitation on the image display and word length, reviewers are recommended to review our response from the complementary material.

---

### Decision · Program_Chairs · 2023-01-20

**Decision:**

Accept: poster

**Justification For Why Not Higher Score:**

Bayes risk is a loss that has a long history. Even in the context of CTC, something similar has been done, so technically, the novelty is limited.

**Justification For Why Not Lower Score:**

The approach itself has merit. The repurposing of the approach is awkward, but is good to have.

**Metareview: Summary, Strengths And Weaknesses:**

This paper proposes a Bayes risk version of CTC for training end-to-end automatic speech recognizers. Usually the purpose of discriminative training is to close the gap between the training loss (log likelihood) and the actual task loss (for example character error rates). This paper repurpose the risk functions for training streaming recognizers, proposing one for emitting the labels early and one for downsampling.

All reviewers unanimously praise the writing and the approach. Some reviewers question the choice of baselines, for example, not choosing naive downsampling to compare against. Some reviewers (including me) feel that the repurposing of Bayes risk is a little awkward, and there are probably better, simpler approaches to solve the problems of concern, for example, coming up with a different loss function that is significantly different from CTC or through architectural changes.

Overall, the paper receives positive reviews.

**Note From Pc:**

if the above contains the word "oral" or "spotlight" please see: "oral" presentation means -> notable-top-5% and "spotlight" means -> notable-top-25%. As stated in our emails, we are disassociating presentation type from AC recommendations